# Thalamocortical dysrhythmia detected by machine learning

Sven Vanneste[1], Jae-Jin Song [2] & Dirk De Ridder[3]

Thalamocortical dysrhythmia (TCD) is a model proposed to explain divergent neurological disorders. It is characterized by a common oscillatory pattern in which resting-state alpha activity is replaced by cross-frequency coupling of low- and high-frequency oscillations. We undertook a data-driven approach using support vector machine learning for analyzing resting-state electroencephalography oscillatory patterns in patients with Parkinson's disease, neuropathic pain, tinnitus, and depression. We show a spectrally equivalent but spatially distinct form of TCD that depends on the specific disorder. However, we also identify brain areas that are common to the pathology of Parkinson's disease, pain, tinnitus, and depression. This study therefore supports the validity of TCD as an oscillatory mechanism underlying diverse neurological disorders.

[1] School of Behavioral and Brain Sciences, The University of Texas at Dallas, Richardson, TX 75080, USA. [2] Department of Otorhinolaryngology-Head and Neck Surgery, Seoul National University Bundang Hospital, Seongnam, 13620, Korea. [3] Unit of Neurosurgery, Department of Surgical Sciences, Dunedin School of Medicine, University of Otago, Dunedin, 9054, New Zealand. Correspondence and requests for materials should be addressed to S.V. (email: sven.vanneste@utdallas.edu) or to J.-J.S. (email: jjsong96@gmail.com)

Specific brain oscillatory behavior characterizes resting-state awake[1] and sleep stages[2] in an evolutionarily preserved way[3], as well as perceptual[4], motor[5], and cognitive states[6]. Furthermore, some brain disorders might feature a specific oscillatory signature known as thalamocortical dysrhythmia (TCD)[7–9]. The original TCD model suggests a common underlying oscillatory mechanism present in specific neurological disorders (i.e., Parkinson's disease, neuropathic pain, tinnitus) as well as neuropsychiatric disorders (i.e., depression)[9]. The original description of TCD proposes that normal resting-state alpha activity (8–12 Hz) slows down to theta frequencies (4–8 Hz) in states of deprived input. This theta activity is further associated with an increase in surrounding beta/gamma (25–50 Hz) activity, which results in persistent cross-frequency coupling between theta and gamma activity[8,9]. The underlying idea is that deprivation leads to a thalamocortical column-specific decrease in information processing, which permits the slowing of resting-state thalamocortical activity from alpha to theta frequencies, as less information needs to be processed[10]. Decreased input also results in a reduction of $GABA_A$-mediated lateral inhibition, inducing gamma activity ($> 30$ Hz) surrounding the deafferented thalamocortical columns[8]. This gamma band activity surrounding theta activity is known as the edge effect[8,9].

Theta oscillations may reflect negative symptoms (depression, hearing loss, hypoesthesia, etc.), while gamma-frequency oscillations conversely reflect positive symptoms (tinnitus, pain, etc.)[8,11]. Negative symptoms, linked to slowed alpha or theta activity, might, therefore, be analogous to what is seen in a sensory-deprived sleep stage[8]. The theta wave acts as a long-range carrier wave[12], hypothetically connecting to a theta oscillation-based memory network. The theta wave then acts as a compensatory mechanism to pull missing information from memory if it cannot be obtained from the environment[7]. Tinnitus, pain, movement, and mood-related information, reflected by high-frequency oscillatory activity such as beta and gamma[13], can be nested on this theta wave by means of cross-frequency coupling.

However, the validity of TCD is a matter of ongoing controversy. Recent interest in cross-frequency coupling in physiological states[6,14,15] might lead to a wider acceptance of TCD as a pathological state[7]. It is, therefore, of interest to verify whether a purely data-driven approach by means of a support vector machine (SVM) can reliably detect TCD. We combine source localized resting-state electroencephalography (EEG) with machine learning in the present study to look for a neurologic and neuropsychiatric signature for tinnitus, neuropathic pain, Parkinson's disease, and depression described using TCD in the seminal paper on the model[9]. We used a region of interest (ROI)-based approach. The initial choice of ROIs was based on a meta-analysis of brain areas involved in the pathophysiology of tinnitus[16]. These include tinnitus-specific areas, such as the auditory cortex, as well as non-specific areas, such as the parahippocampus, dorsal anterior and posterior cingulate cortices, and insular cortex, which are common to tinnitus and the other pathologies[17]. This was complemented by spatially specific areas such as the somatosensory cortex[18], motor cortex[19], and subgenual anterior cingulate cortex[20] that have been associated with neuropathic pain, Parkinson's disease, and depression, respectively. We further aim to establish whether TCD as an entity can be diagnosed using resting-state EEG and further subdivided into its specific clinical entities. Based on theoretical underpinnings[7], we hypothesize that if TCD exists it should be characterized by spectrally equivalent but spatially distinct forms. Our results show a spectrally equivalent but spatially distinct form of TCD depending on the specific neural disorder together with brain areas that are involved in pain, tinnitus, Parkinson's disease, and depression.

## Results

**Whole-brain frequency analysis.** Comparing the power spectra of patients (i.e., tinnitus, pain, Parkinson's disease, and depression) with healthy control subjects showed a significant effect for tinnitus ($F = 4.44$, $p < 0.001$), pain ($F = 7.77$, $p < 0.001$), Parkinson's disease ($F = 3.24$, $p < 0.001$), and depression ($F = 3.29$, $p < 0.001$). A simple contrast analysis showed a significant increase in the current density for tinnitus patients in comparison to healthy control subjects between 2–4 and 14–44 Hz. For pain, in comparison to healthy subjects, we see a significant increase between 2–5 and 14–44 Hz in current density and a significant decrease between 9–10 Hz. For Parkinson's disease patients and patients with depression, we found significant increases from 3–8 and from 3–9 Hz, respectively, in comparison to healthy control subjects. In addition, a significant increase was identified in current density between 12–44 Hz for Parkinson's disease patients and between 19–41 Hz for patients with depression in comparison to healthy control subjects. A general comparison between all patients (i.e., tinnitus, pain, Parkinson's disease, and depression) and healthy control subjects showed a significant effect ($F = 5.07$, $p < 0.001$). A simple contrast analysis revealed a significant increase between 2–5 Hz and between 13–44 Hz. See Fig. 1 for an overview of these results.

**Accuracy and selected cortical areas in each model.** Using SVM learning for tinnitus, we were able to differentiate between tinnitus and healthy control subjects with an average 87.71% (sd = 1.37) accuracy rate in comparison to a random model, which was only 53.30% accurate (sd = 2.66) ($F = 2648.86$, $p < 0.001$). The true-positive ratio (TPR) was on average 0.82 (sd = 0.02) and the false-positive ratio (FPR) was 0.08 (sd = 0.02). In comparison to the random model, the TPR was on average 0.47 (sd = 0.02) ($F = 3138.04$, $p < 0.001$) and the FPR was 0.53 (sd = 0.03) ($F = 5046.92$, $p < 0.001$). The ROC shows a significant effect ($F = 3509.78$, $p < 0.001$), indicating a higher score for the tinnitus model ($M = 0.94$, sd = 0.03) in comparison to the random model ($M = 0.45$, sd = 0.03). A significant difference was also obtained by comparing the κ-statistic (real: $M = 0.75$, sd = 0.02 vs. random: $M = -0.10$, sd = 0.01; $F = 27993.41$, $p < 0.001$), mean average error (MAE) (real: $M = 0.16$, sd = 0.02 vs. random: $M = 0.51$, sd = 0.02; $F = 3822.82$, $p < 0.001$), and root mean squared error (RMSE) (real: $M = 0.26$, sd = 0.02 vs. random: $M = 0.52$, sd = 0.02; $F = 1307.12$, $p < 0.001$) (Fig. 2, uppermost "tinnitus" panel). The model selected the left and right auditory cortices and included the theta, alpha, and gamma-frequency bands, the left parahippocampus at the gamma-frequency band, and the right parahippocampus at the theta and gamma-frequency bands. The model obtained using SVM learning also included the theta, beta, and gamma-frequency bands for the dorsal anterior cingulate cortex and the gamma-frequency band for the subgenual anterior cingulate cortex. In addition, the posterior cingulate cortex for the theta, beta, and gamma-frequency bands and the right insula at the theta frequency contribute to the model (Fig. 3 upper left "tinnitus" panel).

The model obtained using SVM learning for pain was able to differentiate between pain and healthy controls subjects with an average 92.53% (sd = 1.59) accuracy rate, in comparison to a random model which was only 52.74% accurate (sd = 1.79) ($F = 5512.32$, $p < 0.001$). The TPR of the model was on average 0.93 (sd = 0.02) and the FPR was 0.21 (sd = 0.02). In comparison, the random model TPR was on average 0.53 (sd = 0.02) ($F = 4563.30$, $p < 0.001$) and the FPR was 0.45 (sd = 0.03) ($F = 1206.75$, $p < 0.001$). The ROC shows a significant difference between the pain model ($M = 0.95$, sd = 0.01) and the random model ($M = 0.54$, sd = 0.02) ($F = 6317.48$, $p < 0.001$). A

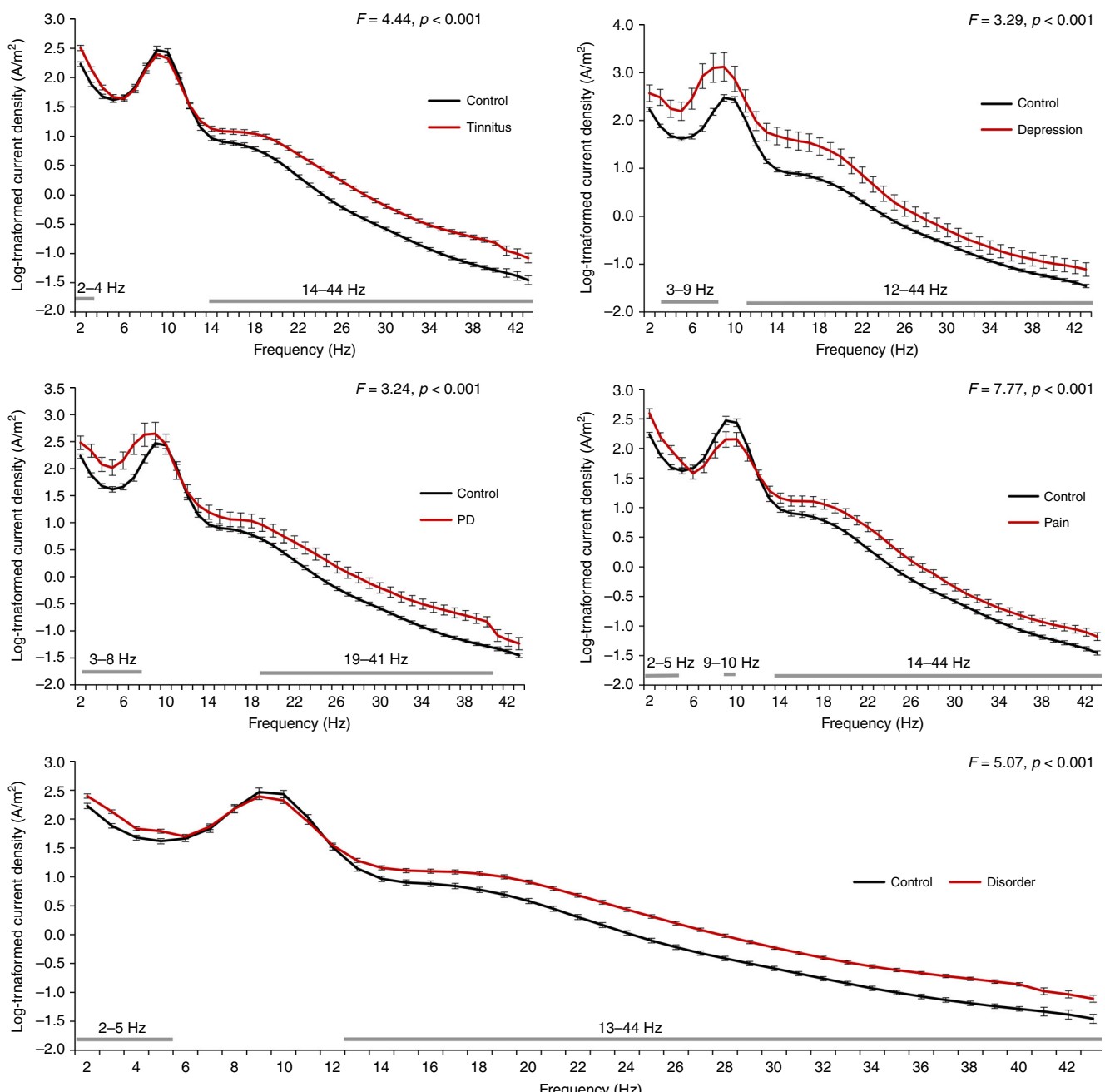

**Fig. 1** A comparison of the power spectrum of patients (i.e., tinnitus ($N = 153$), pain ($N = 78$), PD ($N = 31$), and depression ($N = 15$)) with healthy control subjects ($N = 264$) showed a significant effect for tinnitus ($F = 4.44$, $p < 0.001$), pain ($F = 7.77$, $p < 0.001$) PD ($F = 3.24$, $p < 0.001$), and depression ($F = 3.29$, $p < 0.001$) for specific frequencies (see gray bars in figure). A general comparison between all patients ($N = 277$) (i.e., tinnitus, pain, PD, and depression) and healthy control subjects showed a significant effect ($F = 5.07$, $p < 0.001$) for specific frequencies (see gray bars in figure). Black whiskers indicate standard errors

significant difference was also obtained comparing the κ-statistic (real: $M = 0.78$, sd $= 0.02$ vs. random: $M = 0.07$, sd $= 0.01$; $F = 19779.991$, $p < 0.001$), MAE (real: $M = 0.08$, sd $= 0.02$ vs. random: $M = 0.49$, sd $= 0.02$; $F = 5771.21$, $p < 0.001$), and RMSE (real: $M = 0.21$, sd $= 0.02$ vs. random: $M = 0.51$ sd $= 0.02$; $F = 357.89$, $p < 0.001$) (Fig. 2, the second panel from the top, "pain"). Using SVM learning, the model obtained includes the left and right parahippocampus at the gamma-frequency band, the theta and beta frequency bands for the dorsal anterior cingulate

cortex, and the theta frequency band for the subgenual anterior cingulate cortex. Also, the left insula for the theta frequency band and the right insula for the theta and beta frequency bands were included in the model. The dorsal anterior cingulate cortex was included for the theta and beta frequency bands. The left somatosensory cortex was involved for the gamma-frequency band and the right somatosensory cortex for the theta and alpha frequency bands. In addition, the theta and beta frequency bands

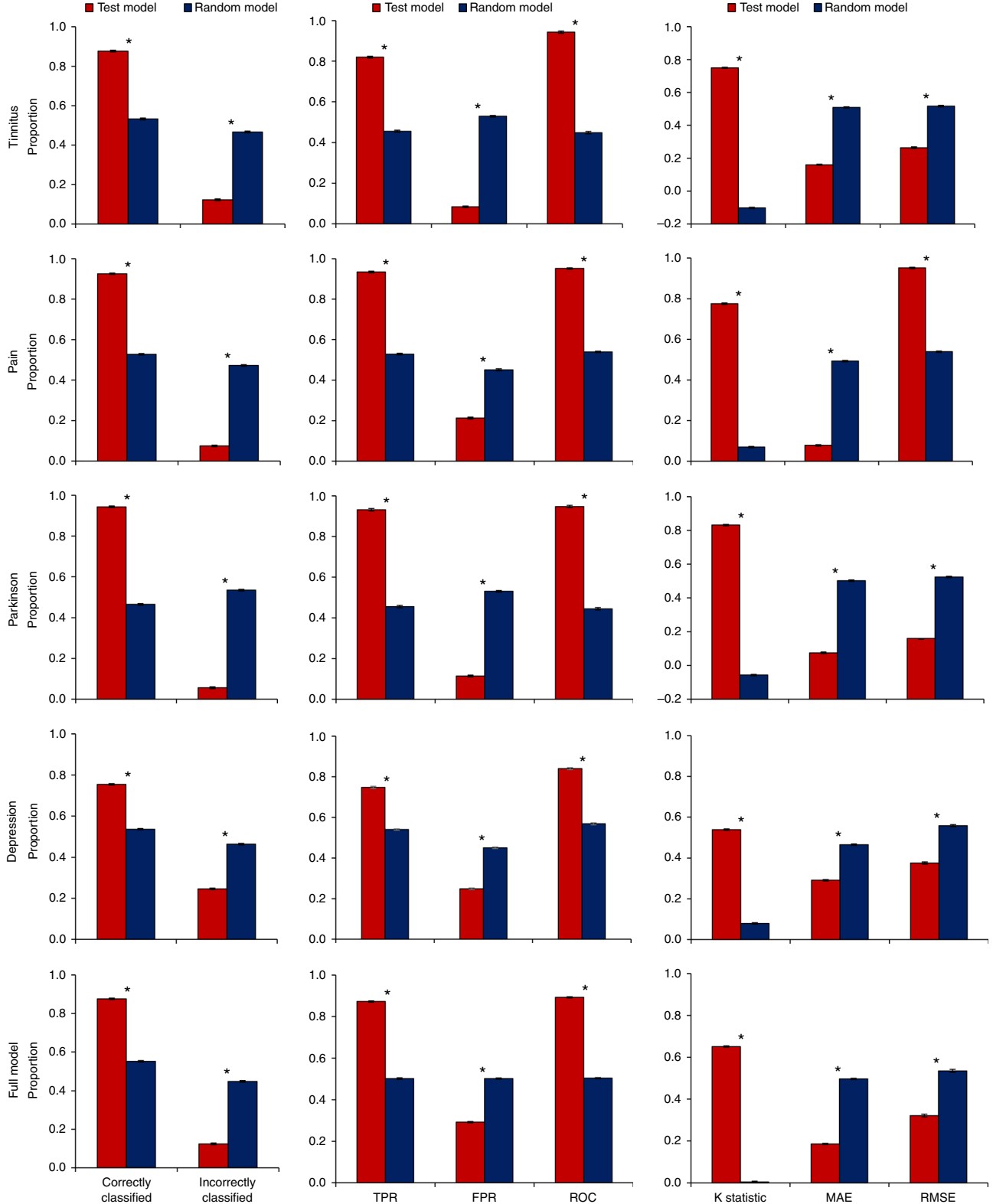

**Fig. 2** Obtained model using support vector machine learning to differentiate between, respectively, tinnitus (*N* = 153) vs. controls (*N* = 264), pain (*N* = 78) vs. controls (*N* = 264), Parkinson disease (*N* = 31) vs. controls (*N* = 264), and depression (*N* = 15) vs. controls (*N* = 264). SVM learning can differentiate between the disorder and healthy control subjects with an accuracy between 75 and 94% in comparison to a random model. The true-positive rate (TPR) of the models and the area under the curve (ROC) were significantly higher for the obtained model in comparison to the random model, while the false-positive rate (FPR) was significantly lower. A significant difference was also identified by comparing the κ-statistic MAE and RMSE, confirming the strength of the tested model in comparison to the random model. (*indicates a significant effect *p* < 0.001). Black whiskers indicate standard errorsPlease check the edits to the sentence 'The true positive rate………' in figure caption 2 is ok.Is correct

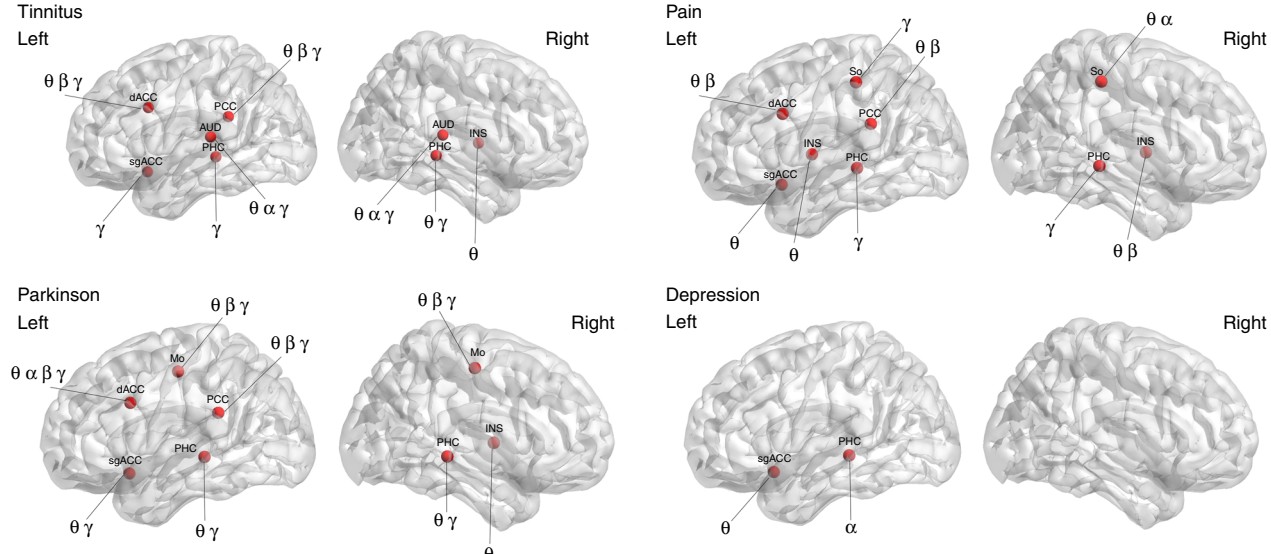

**Fig. 3** Support vector machine learning differentiates between, respectively, tinnitus ($N = 153$) vs. controls ($N = 264$), pain ($N = 78$) vs. controls ($N = 264$), Parkinson disease ($N = 31$) vs. controls ($N = 264$), and depression ($N = 15$) vs. controls ($N = 264$). dACC dorsal anterior cingulate cortex, sgACC subgenual anterior cingulate cortex, INS insula, PHC parahippocampus, AUD auditory cortex, So somatosensory cortex, Mo motor cortex, PCC posterior cingulate cortex, θ theta, α alpha, β beta, γ gamma

for the posterior cingulate cortex were included in the model (Fig. 3, upper right "pain" panel).

For Parkinson's disease, SVM learning obtains a model that classifies on average 94.34% (sd = 1.81) of the Parkinson patients correctly in comparison to a random model 46.56% (sd = 2.17) ($F = 5687.61$, $p < 0.001$). The TPR of the Parkinson model was on average 0.93 (sd = 0.02) and the false-positive rate was 0.11 (sd = 0.01). In comparison to the random model, the TPR was on average 0.46 (sd = 0.02) ($F = 6746.61$, $p < 0.001$) and the FPR was 0.53 (sd = 0.02) ($F = 7267.93$, $p < 0.001$). The ROC shows a significant difference between the Parkinson's model ($M = 0.95$, sd = 0.02) and the random model ($M = 0.45$, sd = 0.02) ($F = 8361.86$, $p < 0.001$). A significant difference was also obtained comparing the κ-statistic (real: $M = 0.83$, sd = 0.02 vs. random: $M = -0.05$, sd = 0.02; $F = 19041.13$, $p < 0.001$), MAE (real: $M = 0.07$, sd = 0.02 vs. random: $M = 0.50$, sd = 0.02; $F = 3928.08$, $p < 0.001$), and RMSE (real: $M = 0.16$, sd = 0.02 vs. random: $M = 0.52$, sd = 0.02; $F = 3563.81$, $p < 0.001$) (Fig. 2, the third panel from the top, "Parkinson"). The model using SVM learning method is selecting the theta and gamma-frequency bands for the left and right parahippocampus; the theta, beta, and gamma-frequency bands for the motor cortex; and the theta, alpha, beta, and gamma-frequency bands for the dorsal anterior cingulate cortex. For the subgenual anterior cingulate cortex, the gamma frequency was selected. In addition, the theta frequency band was selected for the right insula and the theta, beta, and gamma-frequency bands for the posterior cingulate cortex (Fig. 3 lower left "Parkinson" panel).

Our depression model shows it was able to classify on average 75.40% (sd = 1.76) in comparison to a random model at 52.58% (sd = 1.58) ($F = 1699.06$, $p < 0.001$). The TPR of the depression model ($M = 0.75$, sd = 0.02) was significant in comparison to the random model ($M = 0.54$, sd = 0.01) ($F = 1707.00$, $p < 0.001$). The FPR was on average 0.25 (sd = 0.02) for the depression model, while for the random model it was on average 0.45 (sd = 0.01) ($F = 1286.57$, $p < 0.001$). The ROC shows a significant difference between the depression model ($M = 0.84$, sd = 0.02) and the random model ($M = 0.57$, sd = 0.03) ($F = 1410.76$, $p < 0.001$). A significant difference was also obtained comparing the

κ-statistic (real: $M = 0.54$, sd = 0.02 vs. random: $M = 0.08$, sd = 0.02; $F = 8427.24$, $p < 0.001$), MAE (real: $M = 0.29$, sd = 0.02 vs. random: $M = 0.47$, sd = 0.02; $F = 1193.82$, $p < 0.001$), and RMSE (real: $M = 0.38$, sd = 0.03 vs. random: $M = 0.56$, sd = 0.02; $F = 611.02$, $p < 0.001$) (Fig. 2, the second panel from the bottom, "Depression"). Using SVM learning, the model selected the alpha frequency band of the left parahippocampus and the theta frequency band of the subgenual anterior cingulate cortex (Fig. 3 lower right "Depression" panel).

Using the full model (including tinnitus, pain, Parkinson, depression), 87.60% (sd = 1.21) of the subjects were correctly classified in comparison to a random model 55.15% (sd = 2.50) ($F = 2733.02$, $p < 0.001$). The TPR of the model ($M = 0.87$, sd = 0.02) in comparison to a random model ($M = 0.50$, sd = 0.02) was higher ($F = 5606.06$, $p < 0.001$). The FPR was on average 0.29 (sd = 0.01) for the full model, while for the random model the average was 0.50 (sd = 0.01) ($F = 2236.58$, $p < 0.001$). The ROC shows a significant difference between the full TCD model ($M = 0.89$, sd = 0.01) and random model ($M = 0.50$, sd = 0.01) ($F = 9257.55$, $p < 0.001$). A significant difference was also obtained comparing the κ-statistic (real: $M = 0.65$, sd = 0.01 vs. random: $M = 0.003$, sd = 0.02; $F = 1708.83$, $p < 0.001$), MAE (real: $M = 0.19$, sd = 0.01 vs. random: $M = 0.50$, sd = 0.01; $F = 5184.07$, $p < 0.001$), and RMSE (real: $M = 0.32$, sd = 0.04 vs. random: $M = 0.54$, sd = 0.02; $F = 461.73$, $p < 0.001$) (Fig. 1, the lowermost panel, "Full Model"). To differentiate between healthy controls and TCD (including tinnitus, pain, Parkinson's, and depression), the SVM learning model selected the theta, beta, and gamma-frequency bands for the dorsal anterior cingulate cortex; the theta frequency band for the subgenual anterior cingulate cortex; and the gamma-frequency band for the left and right parahippocampus. In addition, the theta and gamma frequencies were selected for the posterior cingulate cortex and the theta frequency band was selected for the insula (Fig. 4).

*Non-TCD group*: Applying the same method to a non-TCD-related pathology (i.e., obesity) shows only 59.96% (sd = 2.42) of the subjects were correctly classified in comparison to a random model 54.95% (sd = 2.31) (n.s.). Additionally, the TPR of the model ($M = 0.52$, sd = 0.11) in comparison to a random model

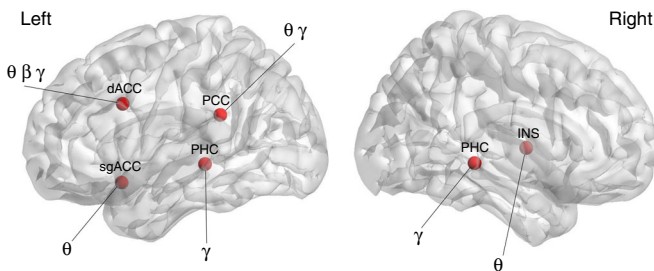

**Fig. 4** Support vector machine learning differentiates between thalamocortical dysrhythmia disorder (*N* = 277) including tinnitus, pain, Parkinson, and depression vs. healthy controls subjects (*N* = 264). dACC dorsal anterior cingulate cortex, sgACC subgenual anterior cingulate cortex, INS insula, PHC parahippocampus, AUD auditory cortex, So somatosensory cortex, Mo motor cortex, PCC posterior cingulate cortex, θ theta, β beta, γ gamma

(*M* = 0.52, sd = 0.09) did not show a significant difference (n.s.). The FPR was on average 0.54 (sd = 0.06) for the test model, while for the random model the average was 0.49 (sd = 0.02) (n.s.). The ROC did not show a significant difference between the non-TCD model (*M* = 0.47, sd = 0.01) and the random model (*M* = 0.48, sd = 0.01). No significant difference was demonstrated by comparing the κ-statistic (real: *M* = 0.07, sd = 0.05 vs. random: *M* = 0.004, sd = 0.06), MAE (real: *M* = 0.49, sd = 0.04 vs. random: *M* = 0.50, sd = 0.01) and RMSE (real: *M* = .54, sd = 0.05 vs. random: *M* = 0.54, sd = 0.03) (see Supplementary Fig. 2).

**Conjunction analysis**. A conjunction analysis between Parkinson's disease, neuropathic pain, tinnitus, and depression revealed a significant effect (*Z* = 2.15, *p* < 0.05) for the dorsal anterior cingulate cortex/midcingulate cortex and parahippocampus for the beta frequency band. No significant effect was obtained for the delta, theta, alpha, or gamma-frequency bands. See Fig. 5.

**Cross-frequency coupling**. A cross-frequency coupling analysis showed significantly increased nesting of theta–beta (*F* = 18.11, *p* < .001) and theta–gamma (*F* = 19.74, *p* < 0.001) for the tinnitus group in the auditory cortex in comparison to pain, Parkinson's disease, depression, and healthy controls. Significantly increased nesting of theta–beta (*F* = 20.21, *p* < 0.001) and theta–gamma (*F* = 17.87, *p* < 0.001) was obtained for the somatosensory cortex for pain patients in comparison to tinnitus patients, Parkinson's disease patients, and patients with depression. For the motor cortex, a significant increase in nesting of theta–beta (*F* = 2.58, *p* = 0.037) and theta–gamma (*F* = 2.92, *p* = 0.021) in patients with pain or Parkinson's disease was revealed in comparison to tinnitus patients, patients with depression, and healthy controls. For both the subgenual anterior cingulate cortex and the dorsal anterior cingulate cortex, significantly increased nesting of theta–beta (*F* = 18.86, *p* < 0.001; *F* = 21.03, *p* < 0.001) and theta–gamma (*F* = 19.51, *p* < 0.001; *F* = 20.67, *p* < 0.001) correlations were found for the tinnitus, pain, Parkinson's disease, and depression patients in comparison to healthy controls. See Fig. 6 for an overview.

The cross-correlation between spectral amplitudes at different frequencies for healthy control subjects as well as for patients with tinnitus, pain, Parkinson's disease, and depression as illustrated in Fig. 7 does not show a significant difference in increased power to power in theta–beta and theta–gamma correlation between the healthy control group and the patient groups.

## Discussion

The objective of this study is to investigate whether there is spectral equivalence between different neurological (i.e., tinnitus, pain, Parkinson's disease) and neuropsychiatric (i.e., depression) disorders with spatially distinct forms of TCD. The results of this study, using a purely data-driven classification by means of SVM learning, clearly show that the theta, beta, and gamma-frequency bands are important in differentiating between neuropsychiatric disorders and healthy control subjects as proposed by and in confirmation of the TCD model. This purely data-driven approach also selected spatially distinct brain areas to discriminate between the different clinical TCD entities. For these spatially distinct brain areas, phase–amplitude cross-frequency coupling was demonstrated between theta–gamma and theta–beta oscillations for tinnitus, pain, Parkinson's disease, and depression. For the dorsal anterior cingulate and subgenual anterior cingulate cortices, an increased coupling between theta–gamma and theta–beta oscillations is identified that is not related to a specific disorder but probably has a more general role (Fig. 8). Theta–beta and theta–gamma coupling were, however, not confirmed when using a power-to-power cross-frequency coupling analysis as applied in the original TCD model[9]. However, more recent research suggests that phase–amplitude coupling more accurately reflects the physiological mechanism for effective communication in the human brain[6].

The data-driven classification method applied to source localized resting-state EEG in different neurological (i.e., tinnitus, pain, Parkinson's disease) and neuropsychiatric (i.e., depression) disorders identifies a temporal pattern of neural oscillations that serves as a cortical signature in accordance with the TCD model. Using SVM learning, the theta, alpha, beta, and gamma-frequency bands were selected in spatially distinct brain areas. These findings were further confirmed doing a whole-brain power spectrum analysis, showing changes in the theta, alpha, beta, and gamma frequencies in comparison to healthy controls. Cross-frequency coupling might be important for integration via low-frequency coherence of distributed, geographically-focal, high-frequency activity[21]. In physiological circumstances, alpha–gamma coupling may be related to perceptual processing via thalamocortical circuits, whereas theta–gamma processing might be related to physiological memory related processing[7]. Pathological theta–gamma coupling would, therefore, be based on slowing of alpha ( = theta)–gamma coupling[22,23]. This further suggests that the theta activity in TCD is actually slowed alpha, as originally proposed[9].

This temporal pattern goes together with a typical spatial pattern of neural activity that is dependent on the neurological or neuropsychiatric disorder. While the somatosensory cortex is important for neuropathic pain, in tinnitus the auditory cortex is a significant contributor, and in Parkinson's disease the motor cortex plays a distinctive role. These findings confirm previous research and suggest that each disorder has a unique cortical signature. However, it is incorrect to conclude based on reverse inference that these patterns of brain activation represent pain, tinnitus, or Parkinson's disease, respectively. In addition, our data demonstrate an increase in theta–beta and theta–gamma coupling for these specific areas related to the specific disorder. For pain, we see increased coupling at the somatosensory cortex; for tinnitus, we see increased coupling at the auditory cortex; and for Parkinson's disease, increased coupling is seen at the motor cortex. Interestingly, for Parkinson's disease we also observe increased coupling in the auditory and somatosensory cortices, but this does not survive correction for multiple comparisons. The motor cortex findings of cross-frequency coupling on EEG are in keeping with recordings from the subthalamic nucleus in

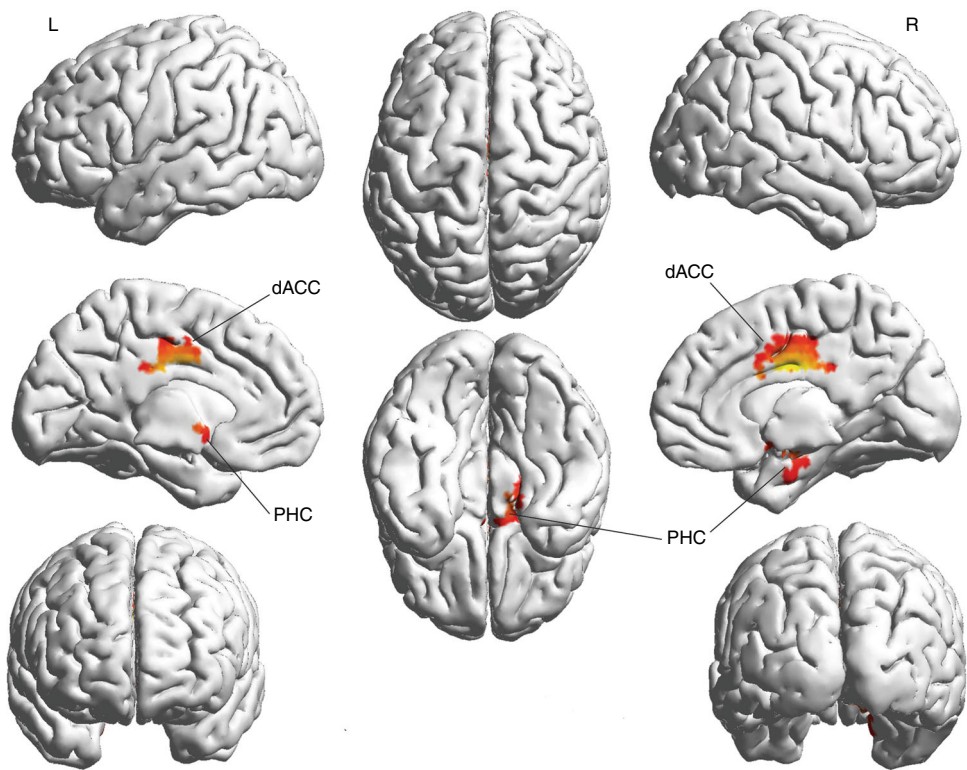

**Fig. 5** Conjunction analysis between tinnitus, pain, Parkinson's, and depression after the subtraction of the healthy controls shows a significant increase in the dorsal anterior cingulate cortex and parahippocampal area for the beta frequency band. dACC dorsal anterior cingulate cortex, PHC parahippocampus

Parkinson's disease, which also shows symptoms related to cross-frequency coupling[24].

For depression, the involvement of the subgenual anterior cingulate cortex is not straightforward, as this area also characterizes the three other disorders in the present study. Convergence of findings across modalities and mood states has identified the subgenual anterior cingulate cortex as a critical brain region in emotion, autonomic nervous system processing, and the pathogenesis of mood disorders[25–27]. This could explain why the subgenual anterior cingulate cortex is a contributor to tinnitus, pain, and Parkinson's disease. It is known that these neurological disorders are associated with mood changes. The fact that our data-driven classification approach identifies the subgenual anterior cingulate cortex as an important brain area to differentiate between healthy controls and specific neurological and neuropsychiatric disorders suggests that it could indeed be involved in common aspects of these disorders. However, we also see a different temporal pattern of neural oscillations for the subgenual anterior cingulate depending on the disorder. For both tinnitus and Parkinson's disease, the gamma wave seems to be important; for pain and depression, the theta wave is more prominent.

The commonality between the different neurological and neuropsychiatric disorders and the dorsal anterior cingulate cortex is not unexpected. The dorsal anterior cingulate cortex together with the anterior insula comprises the core of the salience network and is a key node that overlaps with the psychiatric and neurological "common core" map[28]. It is a critical hub within the functional architecture of the brain at the intersection of cognitive, affective, and somatosensory processing[27]. Animal behavioral studies have further demonstrated that anterior cingulate cortex activation is critical for memory processing involved in long-term negative affective states[29]. Human studies further

show its involvement in tinnitus loudness[30] and distress[17,31] processing, pain processing[32,33], and executive behavior in Parkinson's disease[34]. Hence, it makes sense that this particular area is also involved in tinnitus, pain, Parkinson's disease, and depression. Furthermore, theta–gamma coupling within the dorsal anterior cingulate cortex has been associated with attention and goal directed behavior[35,36].

In this paper, we only look at TCD in the specific neurological (i.e., tinnitus, pain, Parkinson's) and neuropsychiatric (i.e., depression) disorders suggested in the original paper on TCD[9]. Further research has shown that TCD could also be present in schizophrenia[37], migraines[38], visual snow[39], and chronic back pain[40]. Therefore, future research could also look at these additional pathologies and cross-validate our findings by including non-TCD-related disorders. An additional weakness of the paper could be the unequal distribution of the sample sizes of the specific TCDs included in our analysis. Although there is a smaller sample size for Parkinson's disease ($n = 31$) and major depression ($n = 15$), this should not have a major bearing on the individual accuracies of the model, since each pathology is individually compared to a random model and yields approximately the same accuracy. In addition, comparing a weighted with an unweighted model for each disorder revealed approximately similar accuracy rates (see Supplementary Material and Supplementary Fig. 3). Furthermore, the internal validity was confirmed by a tenfold cross-validation technique for each disorder. The unequal distribution of the sample sizes could influence the full model; however, when comparing the expected contribution to the full model based on the sample size (see Supplementary Fig. 4) with the actual contribution based on the accuracy of the model, it is clear that the unequal distribution does not influence the full model.

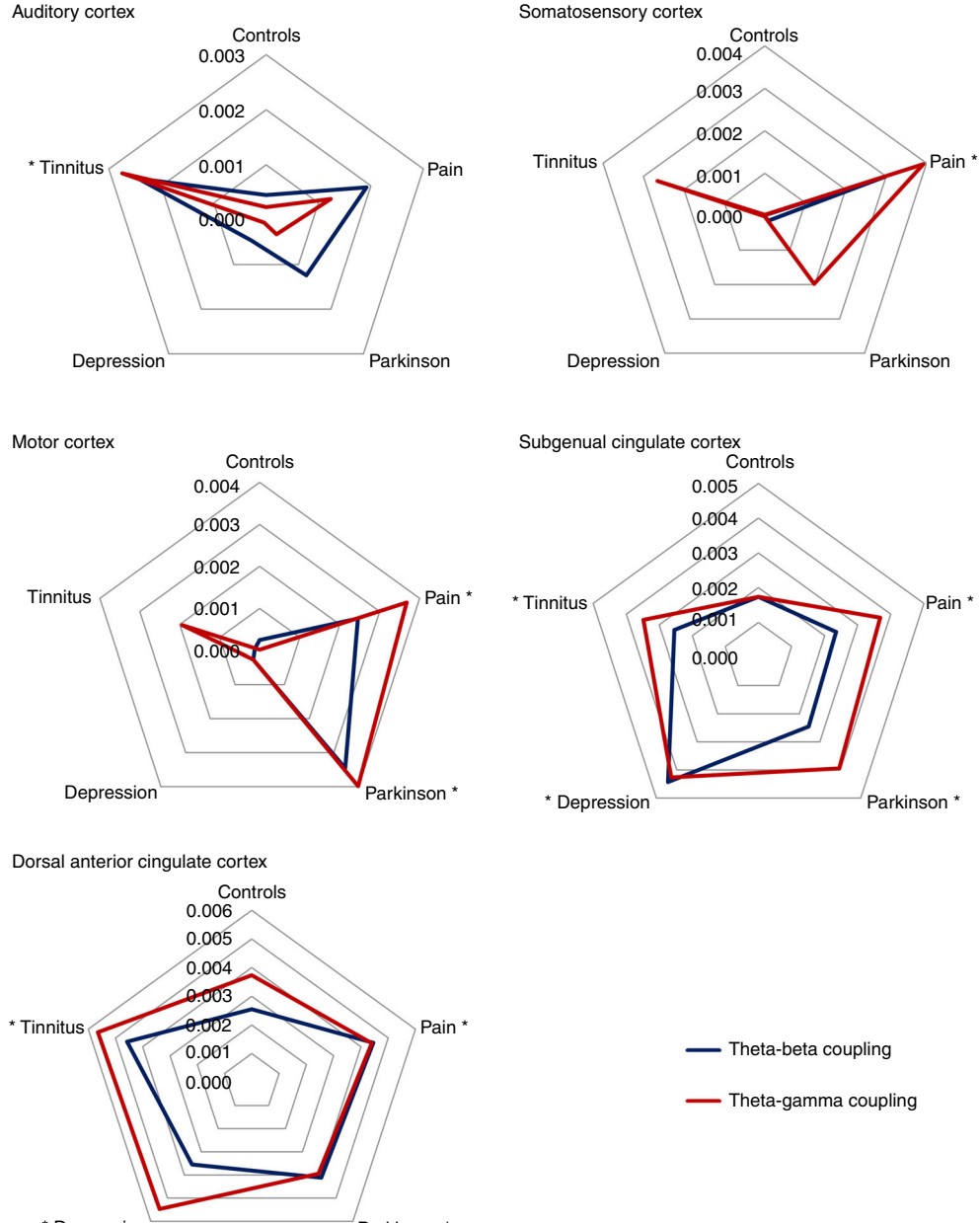

**Fig. 6** Radar plot of presence of cross-frequency coupling in the auditory cortex, somatosensory cortex, motor cortex, subgenual anterior cingulate cortex, and the dorsal anterior cingulate cortex for theta–beta and theta–gamma coupling using Pearson correlations. An asterisk indicates if the CFC for a specific disorder is significantly different in comparison to all other disorders and healthy control subjects after Bonferroni correction ($p < 0.05$). The figure demonstrates the presence of theta–gamma (red line) and theta–beta (black line) coupling for tinnitus in the auditory cortex (upper left panel), for pain in the somatosensory (upper right panel) and motor cortices (mid left panel), and Parkinson's disease in the motor cortex (mid left panel). For the dorsal anterior cingulate (lower left panel) and subgenual anterior cingulate cortices (mid right panel), an increased coupling between theta–gamma and theta–beta oscillations is identified that is not related to the specific neurological/neuropsychiatric disorder, but likely has a more non-specific general role. y-axis represent Pearson correlation r score

In conclusion, the current data-driven approach using machine learning shows temporal and spatial patterns of activity that serve as a cortical signature for, respectively, pain, tinnitus, Parkinson's disease, and depression. Our data suggest a spectrally equivalent but spatially distinct form of TCD depending on the specific neural disorder. However, apart from the disorder-specific spatial signature, common brain areas that are involved in pain, tinnitus, Parkinson's disease, and depression are also identified. Therefore, this study supports the existence of TCD as a mechanism underlying diverse neuropsychiatric disorders. However, more

research is needed to cross-validate these findings, including studies of different neurological and neuropsychiatric disorders.

## Methods

**Participants**. The database for this study consists of 541 subjects (245 women and 296 men; 20–75 years of age, $M = 48.43$; sd $= 12.35$): 264 healthy control subjects, 153 tinnitus subjects, 78 subjects with chronic pain, 31 subjects with Parkinson's disease, and 15 subjects with major depression. The healthy control group reported no history of neurological or neuropsychiatric disorders. Tinnitus subjects were screened by a tinnitus specialist to exclude pulsatile tinnitus, Ménière's disease, otosclerosis, and chronic headache. Neurological disorders such as brain tumors

**Fig. 7** Power to power cross-frequency coupling for each disorder (i.e., tinnitus ($N = 153$), pain ($N = 78$), PD ($N = 31$), and depression ($N = 15$)) and healthy control subjects ($N = 264$). This plot represents the cross-correlation between spectral amplitudes at different frequencies (2–44 Hz). No differences were obtained when comparing the different disorders and the healthy controls

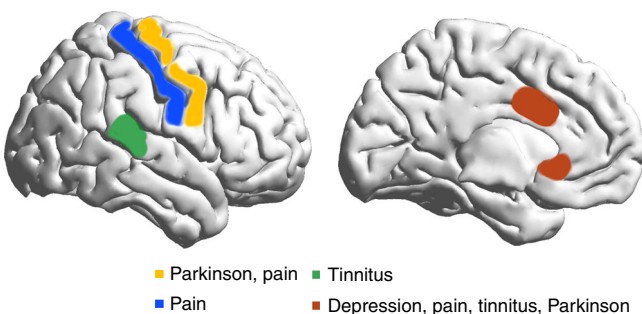

**Fig. 8** Summary figure. Spatial distribution of theta–beta and theta–gamma cross-frequency coupling as related to different thalamocortical dysrhythmia syndromes

were also excluded. All tinnitus patients have had tinnitus for more than 1 year. A pain specialist screened the pain patients for neuropathic pain related to deafferentation (i.e., peripheral nerve, root, or central tract lesions), and that the patients had these pain complaints for >1 year. Parkinson's disease patients were screened by an experienced neurologist who verified all the patients for the following set of criteria: diagnosis of Parkinson's disease according to the UK Brain Bank Criteria, patients in Hoehn & Yahr stage I–III, lack of features suggestive of atypical Parkinsonism, and lack of cognitive dysfunction as evidenced by the Montreal Cognitive Assessment test (MoCA) (score ≥ 26). Patients with major depression for >1 year were screened by an experienced psychiatrist and reported no history of brain injury or lifetime history of primary psychotic ideation, mania,

or substance abuse. All patients with major depression had a score higher than 27 on the Beck Depression Inventory. See Table 1 for further patient characteristics.

The study was in accordance with the ethical standards of the Helsinki declaration (1964) and was approved by the institutional ethics committee at Antwerp University Hospital (UZA OGA85). All participants signed a consent form. All relevant data are available from the authors on request.

**Control non-TCD group.** Forty-five obese participants (31 women and 4 men with a mean age of 49.03, sd = 14.11) were included as a non-TCD control group. Anthropometric measures for the non-TCD group are reported in supplementary table 1. Obesity has been associated with brain changes, but as far as we know, has not been associated with TCD[41]. Hence, we selected this group as a control measure to cross-validate our TCD-related disorders model (i.e., tinnitus, pain, Parkinson's disease, and depression).

**EEG recording.** All resting-state EEGs were recorded in the same room for 5 min at 19 scalp sites of a Tin-electrode cap (ElectroCap, Ohio, United States) using a Mitsar amplifier (Mitsar EEG-201, St. Petersburg, Russia; http://www.mitsar-medical.com) and were sent to the WinEEG software version 2.84.44 (Mitsar, St. Petersburg, Russia). EEGs were measured in a fully lit room shielded from both sound and stray electric currents, with participants sitting upright in a comfortable chair with their eyes closed. The resting-state EEG was sampled with 19 electrodes in the standard 10–20 international placement (Fp1, Fp2, F7, F3, Fz, F4, F8, T7, C3, Cz, C4, T8, P7, P3, Pz, P4, P8, O1, O2), referenced to linked ears, and impedances were maintained below 5 kΩ at all electrodes throughout the EEG recording. Participants were instructed not to drink alcohol for 24 h prior to EEG recording or caffeinated beverages on the day of recording to avoid alcohol- or caffeine-induced changes in the EEG stream[42–44]. The alertness of participants was checked by monitoring both slowing of the alpha rhythm and appearance of spindles in the EEG stream to prevent possible enhancement of the theta power due to drowsiness during recording. No participants included in the current study showed such EEG changes during measurements. Data was recorded in the WinEEG software with a

**Table 1 Patient characteristics**

|  | Healthy (N = 264) | Tinnitus (N = 153) | Pain (N = 78) | Parkinson disease (N = 31) | Depression (N = 15) |
|---|---|---|---|---|---|
| Gender | 152/112 | 79/74 | 43/35 | 17/14 | 5/10 |
| Age | 49.51 (12.54) | 45.42 (12.30) | 47.39 (10.26) | 56.62 (12.32) | 48.51 (13.23) |
| Tinnitus |  |  |  |  |  |
| Type | Lateralization | Tinnitus stress | Loudness | Mean hearing loss |  |
| Pure tone: 61 | Unilateral: 44 | 36.02 (16.32) | 5.02 (2.48) | 31.26 (9.34) |  |
| Narrow band noise: 87 | Bilateral: 109 |  |  |  |  |
| Pain |  |  |  |  |  |
| VAS | PVAQ |  |  |  |  |
| 6.30 (2.01) | 45.21 (10.31) |  |  |  |  |
| Parkinson disease |  |  |  |  |  |
| PDQ | HAM | BDI | UPDRS | PDSS | PSS |
| 80.81 (26.34) | 8.5 (6.32) | 8.47 | 43.44 (15.53) | 74.20 (17.06) | 27.22 (7.01) |
| Depression |  |  |  |  |  |
| BDI |  |  |  |  |  |
| 34.21 (5.34) |  |  |  |  |  |

sampling rate of 1024 Hz, a high-pass filter at 0.15 Hz, and a low-pass filter at 200 Hz. The data were then resampled to 128 Hz, band-pass filtered (fast Fourier transform filter applying a Hanning window) to 2–44 Hz, and imported into the Eureka! Software[45]. A careful inspection of artifacts was performed and all episodic artifacts suggestive of eye blinks, eye movements, jaw tension, teeth clenching, or body movement were manually removed from the EEG stream. An artifact was defined as an EEG characteristic that differs from signals generated by activity in the brain. (1) Some artifacts are known to be in a limited frequency range, e.g., above some frequency. These were removed by frequency filtering. (2) Some artifacts consist of discrete frequencies such as 50 Hz (or 60 Hz for USA) or its harmonics. These were removed by notch filtering. (3) Some artifacts are limited to a certain time range, e.g., in the case of eye blinks. These artifacts were recognized by visual inspection and these time intervals were discarded. (4) Some artifacts originate from one or a few distinct sources or a limited volume of space so that the artifact topography is a superposition of characteristic topographies (equivalently, the artifact is limited to a subspace of the signal space). We removed these artifacts by determining the characteristic topographies (equivalently, the artifact subspace) so that the remaining signals do not contain anything from the artifact subspace. (5) Artifacts and true brain signals that can be assumed to be sufficiently independent can be removed by independent component analysis. (6) Some artifacts are characterized by a particular temporal pattern such as exponential decay. We removed these artifacts by modeling the artifact and fitting its parameters to the data and then removing the artifact.

After artifact rejection, a comparison was made between the different groups (healthy control subjects, tinnitus subjects, subjects with chronic pain, subjects with Parkinson's disease, and subjects with major depression) for the average length of the EEG. This analysis showed no significant differences between the different groups ($F = 0.88$, $p = 0.48$; see Supplementary Fig. 1).

**Source localization analysis**. Standardized low-resolution brain electromagnetic tomography (sLORETA, available at http://www.uzh.ch/keyinst/loreta.htm) is a functional imaging method yielding standardized current density with zero localization error based on certain electrophysiological and neuroanatomical constraints[46]. sLORETA was utilized to estimate the intracerebral sources generating the scalp-recorded electrical activity in each of the following eight frequency bands: delta (2–3.5 Hz), theta (4–7.5 Hz), alpha (8–12 Hz), beta (13–30 Hz), and gamma (30.5–44 Hz)[47]. The sLORETA algorithm solves the inverse problem—the computation of images of electric neuronal activity based on extracranial measurements—by assuming related orientations and strengths of neighboring neuronal sources that are represented by adjacent voxels. The solution space used in this study and associated lead field matrix are those implemented in the LORETA-Key software. This software implements revisited realistic electrode coordinates[48] and the lead field produced by Fuchs et al.[49] by applying the boundary element method on the MNI-152 (Montreal neurological institute, Canada). The sLORETA-key anatomical template divides and labels the neocortical (including the hippocampus and ACC) MNI-152 volume in 6239 voxels with a size of $5 \times 5 \times 5$ mm, based on probabilities returned by the Daemon Atlas (Lancaster et al. 2000)[50]. The co-registration makes use of the correct translation from the MNI-152 space into the Talairach and Tournoux space. Anatomical labeling of significant clusters was done using sLORETA's built-in toolbox. There are concerns that sLORETA analyses are disadvantageous in comparison to functional magnetic resonance imaging (MRI) due to a lower spatial resolution and a restriction to cortical gray matter and hippocampus[51]; however, other studies have validated sLORETA by comparing it with other established localization methods such as structural MRI[52], positron emission tomography[53–55], and functional MRI[56,57]. Further validation of sLOR-ETA has been based on accepting the localization findings obtained from previous invasive studies using implanted electrodes for epilepsy[58,59] and cognitive ERPs[60]

as reasonable evidence. Additionally, previous studies have shown accurate localization of deep brain structures such as the subgenual anterior cingulate cortex[53] and the mesial temporal lobe[61] using sLORETA.

**Region of interest analysis**. The log-transformed electric current density was averaged across all voxels belonging to the regions of interest (ROIs) for the different frequency bands: delta (2–3.5 Hz), theta (4–7.5 Hz), alpha (8–12 Hz), beta (13–30 Hz), and gamma (30.5–44 Hz). The ROIs in the present study are the left and right auditory cortex (BA41), the left and right somatosensory cortex (BAs1, 2, 3), the left and right motor cortex (BA4), the left and right parahippocampus (BA27), the left and right insula (BA13), the dorsal anterior cingulate cortex (BA24), the subgenual anterior cingulate cortex (BA25), and the posterior cingulate cortex (BA23). For the dorsal anterior cingulate cortex, the subgenual anterior cingulate cortex, and the posterior cingulate cortex, we do not differentiate between left and right due to their proximity to the midline. Due to volume conduction, laterality is harder to differentiate for areas close to the midline. The auditory cortex[62], somatosensory cortex[18], motor cortex[19], and subgenual anterior cingulate cortex[20] are areas included in this analyses that have been established in the literature as spatially specific areas, while the parahippocampus, insula, dorsal anterior cingulate cortex, and posterior cingulate cortex have been associated with more general areas in these neural disorders[17,63]. In addition, we calculated the log-transformed electric current density over all 6239 voxels for the tinnitus, pain, Parkinson's disease, depression, and healthy control subjects for the different frequencies from 2 to 44 Hz.

**Model generation**. A support vector machine (SVM) can classify complex data into two classes. The merit of SVMs is to classify data by mapping input vectors into a high- or infinite-dimensional space with some kernel functions and then constructing a hyperplane to separate them into two classes with a possible maximal margin computed. The margin is defined as the distance from the separating hyperplane to the nearest training-data point. The trained model of a SVM classifier can be used to predict to which class an unknown sample belongs. Details on the basic SVM theory can be found elsewhere[64]. Here, we used the SVM program in the data-mining software Weka (Waikato Environment for Knowledge Analysis version 3.7, developed by the University of Waikato Machine Learning Group, available at http://www.cs.waikato.ac.nz/ml/weka/)[65] to perform all classification tasks. The Weka software suite contains a library of algorithms that build predictive models by learning from examples provided in user supplied data sets. We use the default settings as the running parameters. Our data set included for each subject the five frequency bands (i.e., delta, theta, alpha, beta, and gamma) for each ROI (i.e., left and right auditory cortex, the left and right somatosensory cortex, the left and right motor cortex, the left and right parahippocampus, the left and right insula, the dorsal anterior cingulate cortex, the subgenual anterior cingulate cortex, and the posterior cingulate cortex). Thus, the file consisted of the current density for all ROIs for all 541 subjects in the five frequency bands for the model (i.e., Full model). The analysis was also conducted separately for 264 tinnitus subjects, 78 subjects with chronic pain, 31 subjects with Parkinson's disease, and 15 subjects with major depression, each in comparison to healthy control subjects. The criterion for correct classification was defined as when subjects were assigned to the correct group based on the model calculated by the WEKA software (e.g., for the full model: disorder vs. healthy). We used a linear logistic regression-based classifier as the classification method (see Supplementary Material for more information). A tenfold cross-validation was performed on the full data set. Cross-validation is a technique in which the data set is divided into $k$ equal portions called folds. The first fold is used to generate a predictive model of the data set. The data in the remaining $k$—1 folds are then tested against the model,

yielding measurements of model accuracy. A second model is then generated off the second fold, and the remaining $k-1$ folds (which include the fold that created the first model) are tested against this new model again. Subsequently, after all the folds have been used to create and test a model, the average of the values of model accuracy over the $k$-fold cross-validation is presented as the overall accuracy of the model. For example, in a tenfold cross-validation technique for 100 people labeled (disorder vs. healthy), the program takes the 100 labeled data and produces 10 equally sized sets. Each set is further divided into two subgroups: 90 labeled data that are used for training and ten labeled data are used for testing. For the labeled group that is used for training, the program maintains the same distribution. So if out of the 100 people labeled, 60 were healthy subjects and 40 were subjects with a disorder, it will keep this distribution when it selects 90 subjects for training. That would be 54 healthy subjects and 36 subjects with a disorder. It also produces a classifier with an algorithm from 90 labeled data and applies that on the ten testing data for $k_1$. It does the same computation for each of the nine remaining folds ($k_2$–$k_{10}$) and produces nine more classifiers. At the end of a tenfold validation, the average of the ten classifiers produced from ten equally sized sets is calculated and represents the averaged cross-validation. The measurements of model accuracy calculated by the $k$-fold cross-validation technique include the true-positive ratio (TPR), false-positive ratio (FPR), RMSE, MAE, and κ-statistic. The TPR was calculated as the ratio of the total number of correctly classified positive instances (in this case, positive refers to tinnitus patients) over the total number of positive instances in the testing sample. The RMSE is a measure of how well the machine learns the model, and it was calculated by taking the square root of the average of the residuals (errors not explained by the regression equation) over the total sample size. The MAE is simply the average of residuals over the total sample size. The κ-statistic compares the model's observed accuracy with its expected (chance) accuracy by taking the difference in observed and expected accuracy over 1—expected accuracy.

**Randomization of data**. In order to determine significance of the model accuracy for tinnitus/control, pain/control, Parkinson's/control, and depression/control data, averaged model accuracy and statistics were calculated through randomization of the data. This was done by taking the same data set used to generate the tinnitus model and randomly reassigning patient data as either tinnitus (pain, Parkinson's, or depression, respectively) or control. This randomized data set was then used to generate a prediction model and to model accuracy values. This was done 100 times, and the resulting randomized model accuracy statistics were averaged across all trials.

**Conjunction analysis**. We conducted a conjunction analysis with the tinnitus, pain, Parkinson's disease, and depression data after subtracting the healthy controls[66–69]. A conjunction analysis identifies a "common processing component" for two or more tasks/situations by finding areas activated in independent subtractions[66–69]. Friston et al.[67] also indicated that although general conjunction analysis is used in a within-group condition, it can also be applied between groups and was applied in some recent papers[70,71].

**Cross-frequency coupling**. Theta–beta and theta–gamma coupling (e.g., by nesting) are proposed to be an effective manner of communication between cortically distant areas[6]. To verify whether this theta–beta/theta–gamma nesting is present, nesting was calculated for the auditory cortex, the somatosensory cortex, the motor cortex, the subgenual anterior cingulate cortex, and the dorsal anterior cingulate cortex using phase–amplitude cross-frequency coupling. Nesting was computed as follows: first, the time-series for the x, y, and z components of the sLORETA current for each ROI was obtained. Next, these were filtered in the theta (4–7.5 Hz), beta1 (12.5–30 Hz), and gamma (30.5–44 Hz) frequency band-pass regions. These are the time-series of the electrical current in the three orthogonal directions in space. In each frequency band and for each ROI, a principal component analysis was computed and the first component was retained for the theta and gamma bands. The Hilbert transform was then computed on the gamma component and the signal envelope retained. Finally, the Pearson correlation between the theta component and the envelope of the beta/gamma envelope was computed for each individual.

Furthermore, we calculated the correlation plot of the power spectrum of each disorder. This plot was computed by calculating the cross-correlation between spectral amplitudes at different frequencies. These were obtained by computing the multitaper spectrogram and using a moving analysis window, and then computing the correlation coefficient of the two time series, with their means removed. By performing this computation for a two-dimensional grid of points in space, a two-dimensional image of spectral correlations was generated in a similar way to what was suggested in the original TCD paper.

**Statistical analysis**. To compare the power spectra between the different patient groups and the healthy control subjects, we applied a multivariate analysis of variance (MANOVA) with group as the independent variable and the frequency (2–44 Hz) as the dependent variable. Based on a general significant effect, we further tried to disentangle the effect between the patient groups and healthy subjects for each frequency using a univariate analysis of variance (ANOVA).

To compare the different outcome measures (correctly classified, incorrectly classified, TPR, FPR, ROC, κ-statistic, RMSE, and MAE) of the SVM learning approach, we applied a univariate ANOVA with the model (test vs. random) as the independent variable and the outcome measures as the dependent variable. We applied this method for tinnitus, pain, Parkinson's disease, depression, and the full model. To cross-validate our model obtained using SVN learning, we added a non-TCD group and applied a similar method as above.

For the conjunction analysis, we combined the different independent statistical analyses (each disorder vs. control) performed in the same cortical space as the sLORETA images. The two statistical analyses are in the form of sLORETA files containing $z$-scores, i.e., standard Gaussian values. We calculated each voxel and frequency for the conjunction $z$-score. These probabilities correspond to the Gaussian distribution.

For cross-frequency coupling, we applied a univariate ANOVA with the different groups (controls, tinnitus, pain, Parkinson's disease, and depression) as the independent variable and coupling as the dependent variable (theta–beta/theta–gamma coupling). A Bonferroni correction was applied to correct for multiple comparisons to compare the different groups.

Figures 3, 4, 5, and 8 were generated using MATLAB with a graphical user interface, called BrainNet Viewer (www.nitrc.org/projects/bnv/)[72].

**Data availability**. The data sets analyzed during the current study are available from the corresponding author on reasonable request.

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

## Acknowledgements

We would like to thank Jan Ost for helping with data collection and Anusha Mohan for helping with the data processing. This work was supported by the National Research Foundation of Korea (NRF) grant funded by the Korea government (MSIP) (No. 2016R1C1B2007911) and the Seoul National University Bundang Hospital Research Fund 13-2015-010.

## Author contributions

S.V.: data collection, data processing, data-analysis, writing, and revising manuscript; J.-J.S.: data collection, data processing, data-analysis, and writing; D.D.R.: writing and revising manuscript.

## Additional information

**Competing interests:** The authors declare no competing financial interests.

