## [Peer Review File(PDF 865 kb) · Nature Communications]

Reviewers' comments:

Reviewer #1 (Remarks to the Author):

A. Summary of key results

The authors applied a data-driven approach to detect thalamocortical dysrhythmias in a resting-state EEG dataset consisting of healthy controls and subjects with neurological disorders (PD, Depression, Chronic Pain, and Tinnitus). Their SVM-based classification accurately distinguished between disease states and control states by using distinct sets of spectral features (theta, alpha, beta, gamma) from source-localized signals. They showed that beta power increased in the dACC in disease states compared to healthy controls. Finally, they showed that theta-beta and theta-gamma cross-frequency coupling exhibited disease-specific topologies for PD, Chronic Pain, and Tinnitus, in addition to non-specific topologies common to all disease states.

B: Originality and interest

A previous study by Llinas and colleagues (Llinas et al. 1999, PNAS) showed widespread theta power and cross-frequency power spectrum coherence in PD, pain, tinnitus, and depression. The current study supports this study and goes on to obtain an oscillation-based profile of each disease state using the power and cross-frequency coupling as features. Such findings would be of interest to the wider field. However, the limitations of the datasets (mentioned below) make the conclusions unconvincing.

C. Data and methodology.

- It is not clear how consistently the data were collected across subjects and groups. More description of the recording parameters is necessary to clarify this point. For instance, how long were the recordings after artifact removal for each specific group? Differing lengths of signals can introduce bias into the classifier by influencing the feature set.
- The number of PD and depression subjects is very small compared to the controls and other disease states. How might this bias the SVM?
- Please clarify how the folds were obtained. For instance, how many of the 31 PD subjects were in the first fold? How many were in each of the remaining nine folds? If they are evenly distributed, this would suggest that training was done on three to four PD subjects at any given test.
- Cross frequency coupling is not found consistently in PD subjects even when chosen to have similar symptoms (Wang et al., 2016 Neurobiology of Disease). In addition, such coupling is widespread across the cortex, even in healthy subjects (Florin and Baillet 2016 NeuroImage). Given these considerations, one would expect to see a much broader spatial and group distribution than what is illustrated in Figure 5.

D. Appropriate use of statistics and treatment of uncertainties

The work seems reproducible overall. The statistical analysis seems acceptable, but see above for my questions concerning the SVM and cross-validation approach.

E. Conclusions

In terms of disease characterization, the field is moving toward refined classification and subject-specific approaches, since neurological disorders themselves exhibit significant clinical and neurophysiological heterogeneity. The current study proposes the opposite approach by finding common characteristics for each disease state. This is interesting and merits more investigation; however, the study limitations make the overall conclusions unconvincing. Further, while the study may support common cortical oscillations and cross-frequency coupling amongst the four brain disorders investigated, it is certainly a stretch to conclude that the data reported here 'confirms' the TCD model. Supports, possibly, but not confirms.

F. Suggested improvements

- The sample size is fairly uneven (i.e. many controls, and differing numbers of each disease state). Please comment on how this affects the performance of your models?

- How do you address the volume conduction problem when localizing sources in the subgenual cingulate? Although it is a cortical structure, it is fairly deep for a modality like EEG. The authors provide justification using Pizzagalli, et al. 2004 as a reference (Line 140), but there are notable differences in the analysis techniques between this study and the current study (e.g. range of EEG frequencies of interest).
- For training the SVM models, is power the feature that is used for training? Is this computed using the entire five minute recordings (after artifact removal)?
- What is meant by sLORETA having “zero error”?
- Figure 1
 - o Please expand on the description to include the main take-away(s) for the figure.
 - o Consider labeling the y-axis with percent/units to clarify the figure.
- Figure 2 & 3
 - o In figure 2 (PD, left hemisphere) – why is sgACC labeled for theta/gamma on the top and gamma only on the bottom? The text only indicates gamma frequency band. Also for Pain, right hemisphere, why is So labeled with theta/alpha on top and not labeled on bottom? The text indicates the former.
 - o For both figures, presumably the rows are from the same hemisphere only a different perspective, but this is not clearly labeled. In any case, the second row seems redundant given the cortical reconstruction transparency.
 - o Please expand on the caption to clarify that the frequency labels correspond to power in that band. Does this correspond to an increase or decrease?
 - o Consider adding a clarification point in the caption stating that the sgACC, dACC, and PCC do have a left and right component. Presumably this was done due to volume conduction limitations of EEG.
- Figure 4
 - o Please add labels and description to the caption to clarify the measures.
- Figure 5
 - o Typo in labels: “Parkinson”
 - o It would be helpful to include the key points of the figure in the caption.
- Figure 6
 - o Please use a larger font size for the group labels

G: References

Appropriate.

H. Clarity and Context

- Line 166: For clarity and consistency, the ‘1’ in ‘k1’ should be subscripted
- Line 222: Typos/errors.
- Line 252: Typo: “controls subjects”
- Line 292: Grammatical errors

Reviewer #2 (Remarks to the Author):

In this study the authors aim to verify the presence of thalamocortical dysrhythmia (TCD) in various neural disorders using a machine-learning approach. They train a support vector machine model on current density in various frequency bands of resting-state EEG to distinguish between healthy controls and patients with Parkinson’s disease, tinnitus, chronic pain and depression. The authors show that their model is able to distinguish between patients and controls with a high degree of accuracy compared to a randomly generated model, and that the most important predictor oscillations are spatially distinct in the brain for different disorders, but that the dorsal ACC and hippocampal areas appear to be common regions of disturbance across diagnoses. Furthermore, the cross-frequency (theta-gamma and theta-beta) coupling is increased in different brain regions for different disorders. The authors conclude that these findings show strong support

for TCD as a unifying neural mechanism that can also serve as a unique neural signature in diverse neurological and neuropsychiatric disorders.

GENERAL CRITIQUE:

This study takes an interesting and novel machine-learning approach to the important problem of identifying cross-diagnostic biomarkers in neural disorders. However, the justifications for many of the choices made in the study design, such as the disorders the authors considered and the regions of interest selected in the analysis, are only weakly presented in the text. In addition, some key points are unclearly expressed or completely missing, including participant demographical/clinical information and the training and testing of the SVM model, making it difficult to evaluate the soundness of the experimental design and the validity of the conclusions that the authors make. Furthermore, the figures are difficult to interpret and figure captions seem to be incomplete. Therefore I do not recommend the paper for publication in its current form. Additional concerns as well as elaborations on my previous points are provided below.

SPECIFIC COMMENTS & SUGGESTIONS:

- P.3 – The authors do not provide any evidence for why TCD has been proposed as a mechanism in these disorders. The argument for the experiment as currently articulated in the Introduction seems to be only weakly justified. Out of all the possible neurological/neuropsychiatric disorders, why look in particular at PD/tinnitus/MDD/chronic pain? What is so special about thalamocortical oscillations in these diseases? What about other diseases in which thalamocortical information processing is thought to be disrupted, such as schizophrenia? What is the justification for picking these ROIs? Citations to other papers including the Llinás et al. paper are given, but some supporting evidence should be provided in the main text.
- P.4 – The full demographic and symptom information are not provided. It is impossible to evaluate whether control and patient sub-groups are well matched.
- What was done to address possible confounds, such as cognitive performance irrespective of clinical status or duration of illness? Particularly were any patients on medication and if so how was this controlled for?
- Some of the clinical groups are fairly small in size (particularly for depression, which is already the only psychiatric disorder in the sample), which may affect the accuracy and generalizability of the model.
- P. 6 – How accurate was the manual removal of artifacts? A more detailed description of how these artifacts were identified and removed would be useful.
- P. 8 – It is unclear to me how the SVM was trained. In the Model Generation section it reads as though the SVM was given a multi-class dataset but trained only to make binary predictions (tinnitus vs controls), although it is clear from the Results that the model was tested on the other disorders vs controls as well. Or was the model only trained to differentiate between tinnitus and controls, but then used to classify the other disorders vs controls? This point, although critical to evaluating the integrity of the experiment, is ambiguous in the text.
- Is the model able to differentiate between controls and diseases in which TCD is not thought to be involved?
- The authors show that their SVM model identified frequency bands and regions of interest that were significant predictors of group membership. However the nature of the differences in these oscillations is not clear. It would be helpful to visualize these effects in the patient and control groups by plotting their individual power spectra. TCD specifically proposes that neurons are

locked in theta oscillations and so increased theta power should be observed.

- P. 19 – “We show a spectrally equivalent but spatially distinct form of TCD depending on the specific neuropsychiatric disorder.” – I am not sure that this claim is founded. The authors do not show any data on the power spectra of oscillations in the different disorders. This conclusion seems to be based on the result that oscillations in different frequency bands were identified as important predictors in the SVM model, but it appears that different frequencies, as well as different brain regions, are affected in different disorders (for example alpha band does not appear to be in the pain model).

- P. 18 – Drawing conclusions regarding similarities/differences between neurological and neuropsychiatric disorders (such as the subgenual ACC as a “common core”) can be a little ambitious as only three neurological disorders and one neuropsychiatric (with a small sample size) were considered in this study.

- Figure captions should be updated to provide more information (for example in Fig. 5 it is not clear from the figure/caption that the spider plots are showing Pearson correlation coefficients). The authors should also consider labeling the subplots with letters for easier reference.

MINOR POINTS:

- Consider consistently using both “neurological” and “neuropsychiatric” or a more general term such as “neural disorder”.

- There are numerous grammatical mistakes throughout the manuscript – please check carefully.

Reviewer #3 (Remarks to the Author):

This is an impressive manuscript on a large database of patients similar to the original description of TD in PNAS. I am very supportive of its publication since I do think that TD exists and that the controversy has come more from the technical difficulty in demonstrating it. This manuscript does just that. However, I have a major issue that I believe should be addressed.

Since TD is very controversial and this manuscript makes strong claims, I believe that it should not be published without an additional analysis of the EEG similar to the original publication of TD. This analysis should demonstrate that the relevant frequencies are present in the power spectrum of an example patient and the population, show that shifts in frequency are statistically significant and clearly visible in the power spectrum. Another key aspect of TD is the increase in coherence, also illustrated in the original paper, this manuscript must address coherence at the large scale of the EEG.

Response to the reviewers

Reviewer #1

- It is not clear how consistently the data were collected across subjects and groups. More description of the recording parameters is necessary to clarify this point. For instance, how long were the recordings after artifact removal for each specific group? Differing lengths of signals can introduce bias into the classifier by influencing the feature set.

Answer: We understand the concern of the reviewer and therefore made a comparison between the lengths of the EEG after artifact rejection between the different groups. This analysis revealed no significant difference. In addition, we added more clarification about the recording procedure and artifact rejection. The following was added to the manuscript: 'This study was approved by the local ethical committee at Antwerp University Hospital and was in accordance with the Declaration of Helsinki. All participants provided informed consent before the EEG recording. EEG was obtained for each patient as a standard procedure for diagnostic and neuromodulation treatment purposes at the BRAI²N unit in Antwerp University Hospital. All Resting-state EEGs were recorded in the same room for 5 minutes at 19 scalp sites of a Tin-electrode cap (ElectroCap, Ohio, United States) using a Mitsar amplifier (Mitsar EEG-201, St.Petersburg, Russia; <http://www.mitsar-medical.com>) and were sent to the WinEEG software version 2.84.44 (Mitsar, St. Petersburg, Russia). EEGs were measured in a fully lighted room shielded against sound and stray electric currents, with participants sitting upright on a comfortable chair with their eyes closed. The resting-state EEG was sampled with 19 electrodes in the standard 10 – 20 international placement (Fp1, Fp2, F7, F3, Fz, F4, F8, T7, C3, Cz, C4, T8, P7, P3, Pz, P4, P8, O1, O2), referenced to linked ears, and impedances were maintained below 5 k Ω at all electrodes throughout the EEG recording. Participants were instructed not to drink alcohol 24 hours prior to EEG recording or caffeinated beverages on the day of recording to avoid alcohol- or caffeine-induced changes in the EEG stream (1-3). The alertness of participants was checked by monitoring both slowing of the alpha rhythm and appearance of spindles in the EEG stream to prevent possible enhancement of the theta power due to drowsiness during recording (4). No participants included in the current study showed such EEG changes during measurements. Data was recorded in the WinEEG software with a sampling rate of 1024 Hz, a high-pass filter at 0.15 Hz, and a low-pass filter at 200 Hz. The data was then resampled to 128 Hz, band-pass filtered (fast Fourier transform filter applying a Hanning window) to 2–44 Hz, and imported into the Eureka! Software (5). A careful inspection of artifacts was performed and all episodic artifacts suggestive of eye blinks, eye movements, jaw tension, teeth clenching, or body movement were manually removed from the EEG stream. An artifact was defined as an EEG characteristic that differs from signals generated by activity in the brain. 1) Some artifacts are known to be in a limited frequency range, e.g., above some frequency. This was removed by frequency filtering. 2) Some artifacts consist of discrete frequencies such as 50 Hz (or 60 Hz for USA) or its harmonics. These were removed by notch filtering. 3) Some artifacts are limited to a certain time range, e.g., in the case of eye blinks. These artifacts were recognized by visual inspection and these time intervals were discarded. 4) Some artifacts originate from one or a few distinct sources or a limited volume of space so that the artifact topography is a superposition of characteristic topographies (equivalently, the artifact is limited to a subspace of the signal space). We removed the artifacts by determining the characteristic topographies (equivalently, the artifact subspace) so that the remaining signals do not contain anything from the artifact subspace (6). 5) Artifacts and true brain signals that can be assumed to be sufficiently independent can be removed by independent component analysis (7, 8). 6) Some artifacts are characterized by a particular temporal pattern such as exponential decay. We removed these artifacts by modeling the artifact, fitting its parameters to the data, and then removing the artifact (9).

After artifact rejection, a comparison was made between the different groups (healthy control subjects, tinnitus subjects, subjects with chronic pain, subjects with PD, and subjects with major depression) for the average length of the EEG. This analysis showed no significant difference between the different groups ($F = .88$, $p = .48$; see Figure 1S).'

- The number of PD and depression subjects is very small compared to the controls and other disease states. How might this bias the SVM?

Answer: Although the smaller sample for PD subjects and subjects with major depression, we see a clear differentiation between the SVM model obtained with the real data in comparison to the random model with the same data and same distribution. This finding suggests the great sensitivity of the model. Furthermore, the internal validity was confirmed by a 10-fold cross-validation technique for each disorder. Our models are further tested using the k -fold cross-validation technique as well as true-positive ratio, root mean squared error (RMSE), mean average error (MAE), and κ -statistic, further confirming our findings. The unequal distribution of the sample sizes could potentially influence the full model. However, when comparing the expected contribution to the full model based on the sample size (see Figure 2S) with the actual contribution based on the accuracy of the model, it is clear that the unequal distribution does not influence the full model. We added this to the discussion and supplementary material: 'An additional weakness of the paper could be the unequal distribution of the sample sizes of the specific thalamocortical dysrhythmias included in our analysis. Although there is a smaller sample size for PD ($n = 31$) and major depression ($n = 15$), this should not have a major bearing on the individual accuracies of the model, since each pathology is individually compared to a random model and yields approximately the same accuracy. Furthermore, the internal validity was confirmed by a 10-fold cross-validation technique for each disorder. The unequal distribution of the sample sizes could potentially influence the full model. However, when comparing the expected contribution to the full model based on the sample size (see Figure 2S) with the actual contribution based on the accuracy of the model, it is clear that the unequal distribution does not influence the full model.'

- Please clarify how the folds were obtained. For instance, how many of the 31 PD subjects were in the first fold? How many were in each of the remaining nine folds? If they are evenly distributed, this would suggest that training was done on three to four PD subjects at any given test.

Answer: We understand the concern of the reviewer and further explained the method in detail: 'The regions of interest in the different frequency bands were further organized into a file that could be read by the data-mining software Weka (University of Waikato Machine Learning Group, available at <http://www.cs.waikato.ac.nz/ml/weka/>). Thus, the file consisted of the full dataset, which is comprised of the current density for all regions of interest for all 541 subjects in the 5 frequency bands for the model (i.e. Full model). The analysis was also conducted for respectively 264 tinnitus subjects, 78 subjects with chronic pain, 31 subjects with PD, and 15 subjects with major depression in comparison to healthy control subjects separately. Using a simple logistics filter, a 10 fold cross-validation was performed on the full dataset. Cross-validation is a technique in which the data set is divided into k equal portions called folds. The first fold is used to generate a predictive model of the dataset. The data in the remaining $k - 1$ folds are then tested against the model, yielding measurements of model accuracy. A second model is then generated off the second fold, and the remaining $k - 1$ folds (which includes the fold that created the first model) are tested against this new model again. Subsequently, after all the folds have been used to create and test a model, the average of the values of model accuracy over the k -fold cross-validation is presented as the overall accuracy of the model. For example, in a 10-fold cross-validation technique for 100 people labeled (disorder vs. healthy), the program takes

the 100 labeled data and produces 10 equal sized sets. Each set is divided into two subgroups: 90 labeled data that are used for training and 10 labeled data are used for testing. For the labelled group that is used for training, the program maintains the same distribution. So if out of the 100 people labelled, 60 were healthy subjects and 40 were subjects with a disorder, it will keep this distribution when it selects 90 subjects for training. That would be 54 healthy subjects and 36 subjects with a disorder. It also produces a classifier with an algorithm from 90 labeled data and applies that on the 10 testing data for k_1 . It does the same computation for each of the nine remaining folds ($k_2 - k_{10}$) and produces 9 more classifiers. At the end of a 10-fold validation, the average of the 10 classifiers produced from 10 equal sized sets is calculated and represents the averaged cross validation. The measurements for model accuracy calculated by the k -fold cross-validation technique includes the true-positive ratio, root mean squared error (RMSE), mean average error (MAE), and κ -statistic.'

- Cross frequency coupling is not found consistently in PD subjects even when chosen to have similar symptoms (Wang et al., 2016 Neurobiology of Disease). In addition, such coupling is widespread across the cortex, even in healthy subjects (Florin and Baillet 2016 NeuroImage). Given these considerations, one would expect to see a much broader spatial and group distribution than what is illustrated in Figure 5.

Answer: We agree that in PD subjects we see more widespread coupling in the cortex as confirmed by Florin & Baillet (2015); however, in this study, we are only looking at the regions that were important based on the classifier. A closer look at the data in figure 5 also shows that for Parkinson's, we see increased coupling in most areas (AUD, SOM, MOTOR, SgACC, dACC). However, this effect did not survive correction for multiple comparisons. We added this to the paper: 'Interestingly, for PD we also observe increased coupling in the auditory and somatosensory cortices; however, this effect does not survive correction for multiple comparisons. The motor cortex findings of cross-frequency coupling in the EEG are in keeping with recordings from the subthalamic nucleus in PD, which also shows symptoms related to cross frequency coupling (10).'

D. Appropriate use of statistics and treatment of uncertainties

The work seems reproducible overall. The statistical analysis seems acceptable; see above for my questions concerning the SVM and cross-validation approach.

We added this to the paper (see above)

E. Conclusions

In terms of disease characterization, the field is moving toward refined classification and subject-specific approaches, since neurological disorders themselves exhibit significant clinical and neurophysiological heterogeneity. The current study proposes the opposite approach by finding common characteristics for each disease state. This is interesting and merits more investigation; however, the study limitations make the overall conclusions unconvincing. Further, while the study may support common cortical oscillations and cross-frequency coupling amongst the four brain disorders investigated, it is certainly a stretch to conclude that the data reported here 'confirms' the TCD model. Supports, possibly, but not confirms.

Answer: We understand the concern and toned down the conclusion: 'In conclusion, the current data-driven approach using machine learning shows temporal and spatial patterns of activity that serve as a cortical signature for respectively pain, tinnitus, PD, and depression. Our data suggest a spectrally

equivalent but spatially distinct form of TCD depending on the specific neural disorder. However, apart from the disorder-specific spatial signature, common brain areas that are involved in pain, tinnitus, Parkinson's, and depression are identified as well. Therefore, this study supports the existence of TCD as a unifying mechanism underlying diverse neuropsychiatric disorders; however, more research is needed to cross validate these findings, including different neurological and neuropsychiatric disorders.'

F. Suggested improvements

- The sample size is fairly uneven (i.e. many controls, and differing numbers of each disease state). Please comment on how this affects the performance of your models?

Answer: The uneven number could have a direct effect on the power of the classifier. We see that the smallest sample (i.e. the depression group) has less predictive power in comparison to the other pathologies. However, Pain and Parkinson's have smaller samples in comparison to the tinnitus group, but generate a stronger model in generating a classifier. In addition, we see areas involved in depression as confirmed by previous research, thereby cross validating our findings. It is however possible that increasing the sample would generate more areas selected by the classifier that could lead to more accurate model.

'An additional weakness of the paper could be the unequal distribution of the sample sizes of the specific thalamocortical dysrhythmias included in our analysis. Although there is a smaller sample size for PD (n =31) and major depression (n = 15), this should not have a major bearing on the individual accuracies of the model, since each pathology is individually compared to a random model and yields approximately the same accuracy. Furthermore, the internal validity was confirmed by a 10-fold cross-validation technique for each disorder. The unequal distribution of the sample sizes could potentially influence the full model. However, when comparing the expected contribution to the full model based on the sample size (see Figure 2S) with the actual contribution based on the accuracy of the model, it is clear that the unequal distribution does not influence the full model.'

- How do you address the volume conduction problem when localizing sources in the subgenual cingulate? Although it is a cortical structure, it is fairly deep for a modality like EEG. The authors provide justification using Pizzagalli, et al. 2004 as a reference (Line 140), but there are notable differences in the analysis techniques between this study and the current study (e.g. range of EEG frequencies of interest).

Answer: We understand the concern. However previous research by our group and other groups showed the localization of the sgACC in mood disorders (Jawoska et al. 2012; Albert et al. 2012). In addition, the sgACC has been cross validated using different techniques such as EEG and rsfMRI (Grecius et al, 2007; Mayberg et al 2005).

- For training the SVM models, is power the feature that is used for training? Is this computed using the entire five minute recordings (after artifact removal)?

Answer: The length of the EEG is 4 minutes on average. There was no significant difference in length between the different groups. It is important to state that the artifact removal was done by the same person. We added a figure in the supplementary material and text into the paper: 'After artifact rejection a comparison was made between the different groups (healthy control subjects, tinnitus subjects, subjects with chronic pain, subjects with PD, and subjects with major depression) for the average length of the EEG. This analysis showed no significant difference between the different groups ($F = .88, p = .48$; see Figure 1S).'

- What is meant by sLORETA having “zero error”?

Answer: sLORETA is similar to the Dale et al. method: it employs the current density estimate given by the minimum norm solution, and localization inference is based on standardized values of the current density estimates. However, standardization in sLORETA takes a completely different route. In all noise free simulations, only sLORETA has exact, zero error localization. In all noisy simulations, sLORETA has by far the lowest localization errors. In most cases, the spatial spread (i.e. “blurring”) of sLORETA is smaller than that of the Dale method. The consequence is that, unlike the Dale et al. (6) method, sLORETA is capable of exact (zero-error) localization.

- Figure 1

o Please expand on the description to include the main take-away(s) for the figure.

Answer: We added more information. Figure 1 became Figure 2 in the new draft. ‘Figure 2. Obtained model using support vector machine learning to differentiate between respectively tinnitus vs controls, pain vs controls, Parkinson disease vs controls, and depression vs controls. SVM learning can differentiate between the disorder and healthy control subjects with an accuracy between 75% and 94% in comparison to a random model. The sensitivity of the models and the area under the curve were significantly higher for the obtained model in comparison to the random model, while the false discovery rate was significantly lower. A significant difference was also identified by comparing the κ -statistic MAE and RMSE, confirming the strength of the tested model in comparison to the random model’.

o Consider labeling the y-axis with percent/units to clarify the figure.

Answer: We added this to the paper.

- Figure 2 & 3

In figure 2 (PD, left hemisphere) – why is sgACC labeled for theta/gamma on the top and gamma only on the bottom? The text only indicates gamma frequency band. Also for Pain, right hemisphere, why is So labeled with theta/alpha on top and not labeled on bottom? The text indicates the former.

o For both figures, presumably the rows are from the same hemisphere only a different perspective, but this is not clearly labeled. In any case, the second row seems redundant given the cortical reconstruction transparency. Please expand on the caption to clarify that the frequency labels correspond to power in that band. Does this correspond to an increase or decrease? Consider adding a clarification point in the caption stating that the sgACC, dACC, and PCC do have a left and right component. Presumably this was done due to volume conduction limitations of EEG.

Answer: The figures have changed to figure 3 and 4 in the new version. We added this to the paper: ‘Figure 3. Support vector machine learning differentiates between respectively tinnitus vs controls, pain vs controls, Parkinson disease vs controls, and depression vs controls. Abbreviations: dACC: dorsal anterior cingulate cortex; sgACC: subgenual anterior cingulate cortex; INS: Insula; PHC: parahippocampus; AUD: Auditory Cortex, So: Somatosensory cortex; Mo: Motor cortex, PCC: posterior cingulate cortex.

Figure 4. Support vector machine learning differentiates between thalamocortical dysrhythmia as a unifying disorder (including tinnitus, pain, Parkinson’s, and depression) vs controls. Abbreviations: dACC: dorsal anterior cingulate cortex; sgACC: subgenual anterior cingulate cortex; INS: Insula; PHC:

parahippocampus; AUD: Auditory Cortex, So: Somatosensory cortex; Mo: Motor cortex, PCC: posterior cingulate cortex.'

- Figure 4
- o Please add labels and description to the caption to clarify the measures.

Answer: We added this to the paper.

- Figure 5
- Typo in labels: "Parkinson". It would be helpful to include the key points of the figure in the caption.

Answer: We changed this and add more information: 'Figure 6. Radar plot of presence of cross-frequency coupling in the auditory cortex, somatosensory cortex, motor cortex, subgenual anterior cingulate cortex, and the dorsal anterior cingulate cortex for theta-beta and theta-gamma coupling. An asterisk indicates if the CFC for a specific disorder is significant after Bonferroni correction. The figure demonstrates the presence of theta-gamma (red line) and theta-beta (black line) coupling for tinnitus in the auditory cortex (upper left panel), for pain in the somatosensory (upper right panel) and motor cortices (mid left panel), and Parkinson's in the motor cortex (mid left panel). For the dorsal anterior cingulate (lower left panel) and subgenual anterior cingulate cortices (mid right panel), an increased coupling between theta-gamma and theta-beta oscillations is identified that is not related to the specific neurological/neuropsychiatric disorder, but likely has a more non-specific general role.'

- Figure 6
- o Please use a larger font size for the group labels

Answer: We modified this.

H. Clarity and Context

- Line 166: For clarity and consistency, the '1' in 'k1' should be subscripted
- Line 222: Typos/errors.
- Line 252: Typo: "controls subjects"
- Line 292: Grammatical errors

Answer: We modified these typos

Reviewer #2

- P.3 – The authors do not provide any evidence for why TCD has been proposed as a mechanism in these disorders. The argument for the experiment as currently articulated in the Introduction seems to be only weakly justified. Out of all the possible neurological/neuropsychiatric disorders, why look in particular at PD/tinnitus/MDD/chronic pain? What is so special about thalamocortical oscillations in these diseases? What about other diseases in which thalamocortical information processing is thought to be disrupted, such as schizophrenia? What is the justification for picking these ROIs? Citations to other papers including the Llinás et al. paper are given, but some supporting evidence should be provided in the main text.

Answer: We understand the concern of the reviewer and have added additional information to the paper: 'Specific brain oscillatory behavior characterizes resting-state awake (11) and sleep stages (12) in

an evolutionary preserved way (13), as well as perceptual (14, 15), motor (16), and cognitive states (17). Furthermore, some brain disorders might harbor a specific oscillatory signature, known as thalamocortical dysrhythmia (TCD) (18-20). The original TCD model suggests a common underlying oscillatory mechanism present in specific neurological disorders (i.e. Parkinson's disease (PD), neuropathic pain, and tinnitus) as well as neuropsychiatric disorders (i.e. depression) (20). The original description of TCD proposes that normal resting-state alpha activity (8 – 12 Hz) slows down to theta (4 – 8 Hz) activity in states of deprived input and that this theta activity is associated with an increase in surrounding beta/gamma (25 – 50 Hz) activity, which results in persistent cross-frequency coupling between theta and gamma activity (19, 20). The underlying idea is that deprivation leads to a thalamocortical column-specific decrease in information processing, which permits slowing down of resting-state thalamocortical activity from alpha to theta, as less information needs to be processed (21). Decreased input also results in a reduction of GABA_A mediated lateral inhibition, inducing gamma (>30 Hz) band activity surrounding the deafferented thalamocortical columns (19). This gamma band activity surrounding theta activity is known as the edge effect (19, 20).'

'However, whether TCD really exists is controversial and the entity is not widely accepted. Recent interest in cross-frequency coupling in physiological states (17, 22, 23) might lead to a wider acceptance of TCD as a pathological state (18). It is therefore of interest to verify whether a purely data-driven approach by means of a support vector machine (SVM) can reliably detect this entity. This would serve as proof that TCD as a pathological state indeed does exist. In this study we therefore combine source localized resting-state EEG with machine learning to look for a brain-based neurologic and neuropsychiatric signature for tinnitus, neuropathic pain, PD, depression, and all pathologies described as TCDs in the seminal paper on the model (20). We used a ROI based approach. The initial choice of the ROIs was based on a meta-analysis of brain areas involved in the pathophysiology of tinnitus (24). These include tinnitus specific areas such as the auditory cortex and non-specific areas, such as the parahippocampus, dorsal anterior and posterior cingulate cortex, and insula, which are common to tinnitus and the other pathologies (25, 26). This was complemented by spatially specific areas such as the somatosensory cortex (27), motor cortex (28), and subgenual anterior cingulate cortex (29) that have been associated with neuropathic pain, PD, and depression respectively. We further aim to establish whether TCD as an entity can be diagnosed from resting-state electroencephalography (EEG) and further subdivided into its specific clinical entities. Based on theoretical underpinnings (18), we hypothesize that if TCD exists, it should be characterized by different spectrally equivalent but spatially distinct forms of TCD.'

'In this paper, we only look at TCD in specific neurological (i.e. tinnitus, pain, Parkinson's) and neuropsychiatric (i.e. depression) disorders as suggested in the original paper on TCD. Further research has shown that TCD could also be present in schizophrenia (30), migraines (31), visual snow (32), and chronic back pain (33). Therefore, in future research we could also look at these additional pathologies and cross-validate our findings by including non-TCD related disorders'

'In conclusion, the current data-driven approach using machine learning shows temporal and spatial patterns of activity that serve as a cortical signature for pain, tinnitus, PD, and depression respectively. Our data suggest a spectrally equivalent but spatially distinct form of TCD depending on the specific neural disorder. However, apart from the disorder-specific spatial signature, common brain areas that are involved in pain, tinnitus, Parkinson's, and depression are identified as well. Therefore, this study supports the existence of TCD as a unifying mechanism underlying diverse neuropsychiatric disorders. However, more research is needed to cross validate these findings, including different neurological and neuropsychiatric disorders.'

- P.4 – The full demographic and symptom information are not provided. It is impossible to evaluate whether control and patient sub-groups are well matched. What was done to address possible

confounds, such as cognitive performance irrespective of clinical status or duration of illness? Particularly were any patients on medication and if so how was this controlled for?

Answer: We added more information about the different patient groups in table 1. All patients were in a chronic stage of their disease and if patients were taking medication, use was stable for at least 3 months before the EEG recording. Medication was also different over different patients within one group and between groups. In addition, our ethical board did not allow us to stop patients' medication regimens to record an EEG. Patients were not tested for their cognitive performance. However, the brain areas of interest included in our analysis are not directly related to cognitive performance with the exception of the parahippocampus. However, the parahippocampus has also been shown to be associated with aversion (34) and context processing (35), so it is related to more general area rather than a brain-specific area that was not an essential component to differentiate between the different pathologies.

- Some of the clinical groups are fairly small in size (particularly for depression, which is already the only psychiatric disorder in the sample), which may affect the accuracy and generalizability of the model.

Answer: The uneven number can have a direct effect on the power of the classifier. We see that the smallest sample (i.e. the depression group) has less predictive power in comparison to the other pathologies. However, Pain and Parkinson have smaller samples in comparison to the tinnitus group, but generate a stronger model in generating a classifier. In addition, we see the same areas involved in depression as confirmed by previous research, cross validating our findings. It is nonetheless possible that increasing the sample would generate more areas selected by the classifier that could lead to more accurate model. However, when comparing the expected contribution to the full model based on the sample size (see Figure 2S) with the actual contribution based on the accuracy of the model, it is clear that the unequal distribution does not influence the full model. We added this to the discussion and supplementary material: 'An additional weakness of the paper could be the unequal distribution of the sample sizes of the specific thalamocortical dysrhythmias included in our analysis. Although there is a smaller sample size for PD (n =31) and major depression (n = 15), this should not have a major bearing on the individual accuracies of the model, since each pathology is individually compared to a random model and yields approximately the same accuracy. Furthermore, the internal validity was confirmed by a 10-fold cross-validation technique for each disorder. The unequal distribution of the sample sizes could potentially influence the full model. However, when comparing the expected contribution to the full model based on the sample size (see Figure 2S) with the actual contribution based on the accuracy of the model, it is clear that the unequal distribution does not influence the full model.'

- P. 6 – How accurate was the manual removal of artifacts? A more detailed description of how these artifacts were identified and removed would be useful.

Answer: We further clarified this in the paper: 'An artifact was defined as an EEG characteristic that differs from signals generated by activity in the brain. 1) Some artifacts are known to be in a limited frequency range, e.g., above some frequency. This was removed by frequency filtering. 2) Some artifacts consist of discrete frequencies such as 50 Hz (or 60 Hz for USA) or its harmonics. These were removed by notch filtering. 3) Some artifacts are limited to a certain time range, e.g., in the case of eye blinks. These artifacts were recognized by visual inspection and these time intervals were discarded. 4) Some artifacts originate from one or a few distinct sources or a limited volume of space so that the artifact topography is a superposition of characteristic topographies (equivalently, the artifact is limited to a subspace of the signal space). We removed the artifacts by determining the characteristic topographies (equivalently, the artifact subspace) so that the remaining signals do not contain anything from the artifact subspace

(6). 5) Artifacts and true brain signals that can be assumed to be sufficiently independent can be removed by independent component analysis (7, 8). 6) Some artifacts are characterized by a particular temporal pattern such as exponential decay. We removed these artifacts by modeling the artifact and fitting its parameters to the data and then removing the artifact (9).

After artifact rejection, a comparison was made between the different groups (healthy control subjects, tinnitus subjects, subjects with chronic pain, subjects with PD, and subjects with major depression) for the average length of the EEG. This analysis showed no significant difference between the different groups ($F = .88, p = .48$; see Figure 1S).'

- P. 8 – It is unclear to me how the SVM was trained. In the Model Generation section it reads as though the SVM was given a multi-class dataset but trained only to make binary predictions (tinnitus vs controls), although it is clear from the Results that the model was tested on the other disorders vs controls as well. Or was the model only trained to differentiate between tinnitus and controls, but then used to classify the other disorders vs controls? This point, although critical to evaluating the integrity of the experiment, is ambiguous in the text.

Answer: 'The regions of interest in the different frequency bands were further organized into a file that could be read by the data-mining software Weka (University of Waikato Machine Learning Group, available at <http://www.cs.waikato.ac.nz/ml/weka/>). Thus, the file consisted of the full dataset, which is comprised of the current density for all regions of interest for all 541 subjects in the 5 frequency bands for the model (i.e. Full model). The analysis was also conducted for respectively 264 tinnitus subjects, 78 subjects with chronic pain, 31 subjects with PD, and 15 subjects with major depression in comparison to healthy control subjects separately. Using a simple logistics filter, a 10 fold cross-validation was performed on the full dataset. Cross-validation is a technique in which the data set is divided into k equal portions called folds. The first fold is used to generate a predictive model of the dataset. The data in the remaining $k - 1$ folds are then tested against the model, yielding measurements of model accuracy. A second model is then generated off the second fold, and the remaining $k - 1$ folds (which includes the fold that created the first model) are tested against this new model again. Subsequently, after all the folds have been used to create and test a model, the average of the values of model accuracy over the k -fold cross-validation is presented as the overall accuracy of the model. For example, in a 10-fold cross-validation technique for 100 people labeled (disorder vs. healthy), the program takes the 100 labeled data and produces 10 equal sized sets. Each set is divided into two subgroups: 90 labeled data that are used for training and 10 labeled data are used for testing. For the labelled group that is used for training, the program maintains the same distribution. So if out of the 100 people labeled, 60 were healthy subjects and 40 were subjects with a disorder, it will keep this distribution when it selects 90 subjects for training. That would be 54 healthy subjects and 36 subjects with a disorder. It also produces a classifier with an algorithm from 90 labeled data and applies that on the 10 testing data for k_1 . It does the same computation for each of the nine remaining folds ($k_2 - k_{10}$) and produces 9 more classifiers. At the end of a 10-fold validation, the average of the 10 classifiers produced from 10 equal sized sets is calculated and represents the averaged cross validation. The measurements of model accuracy calculated by the k -fold cross-validation technique includes the true-positive ratio, root mean squared error (RMSE), mean average error (MAE), and κ -statistic. True-positive ratio was calculated as the ratio of the total number of correctly classified positive instances (in this case, positive refers to tinnitus patients) over the total number of positive instances in the testing sample. The RMSE is a measure of how well the model is learned by the machine, and was calculated by taking the square root of the average of the residuals (errors not explained by the regression equation) over the total sample size. The MAE is simply the average of residuals over the total sample size. The κ -statistic

compares the model's observed accuracy with its expected (chance) accuracy by taking the difference in observed and expected accuracy over $1 - \text{expected accuracy}$.'

- Is the model able to differentiate between controls and diseases in which TCD is not thought to be involved?

Answer: We added an additional analysis trying to differentiate between a control of a non-TCD described group (i.e. obese patients) and healthy subjects using the exact same model and same regions of interest. This model was not able to differentiate between the disease state and the control group. This suggests that the model is specifically looking at a pattern (increased theta and gamma power) that cannot be detected in non-TCD describe group. This further cross validates our findings.

- The authors show that their SVM model identified frequency bands and regions of interest that were significant predictors of group membership. However the nature of the differences in these oscillations is not clear. It would be helpful to visualize these effects in the patient and control groups by plotting their individual power spectra. TCD specifically proposes that neurons are locked in theta oscillations and so increased theta power should be observed. P. 19 – "We show a spectrally equivalent but spatially distinct form of TCD depending on the specific neuropsychiatric disorder." – I am not sure that this claim is founded. The authors do not show any data on the power spectra of oscillations in the different disorders. This conclusion seems to be based on the result that oscillations in different frequency bands were identified as important predictors in the SVM model, but it appears that different frequencies, as well as different brain regions, are affected in different disorders (for example alpha band does not appear to be in the pain model).

Answer: We understand the concern of the reviewers and added a specific section in the results section dealing with this issue: '*Whole brain Frequency analysis*

Comparing the power spectrum of patients (i.e. tinnitus, pain, PD, and depression) with healthy control subjects showed a significant effect for tinnitus ($F = 4.44, p < .001$), pain ($F = 7.77, p < .001$) ($F = 3.29, p < .001$), PD ($F = 3.24, p < .001$), and depression ($F = 3.29, p < .001$). A simple contrast analysis showed a significant increase in the current density for tinnitus patients in comparison to healthy control subjects between 2-4 Hz and 14-44 Hz. In comparison to healthy subjects, we see for pain a significant increase between 2 to 5 Hz and 14 to 44 Hz in current density and a significant decrease between 9 to 10 Hz. For both PD patients and patients with depression, we found respectively a significant increase from 3 to 8 Hz and from 3 and 9 Hz in comparison to healthy control subjects. In addition, a significant increase was identified in current density between 12 and 44 Hz for PD patients and between 19 and 41 Hz for patients with depression in comparison to healthy control subjects. A general comparison between all patients (i.e. tinnitus, pain, PD, and depression) and healthy control subjects showed a significant effect ($F = 5.07, p < .001$). A simple contrast analysis revealed a significant increase between 2 and 5 Hz and between 13 and 44 Hz. See Figure 1 for overview.'

- P. 18 – Drawing conclusions regarding similarities/differences between neurological and neuropsychiatric disorders (such as the subgenual ACC as a "common core") can be a little ambitious as only three neurological disorders and one neuropsychiatric (with a small sample size) were considered in this study.

Answer: We tone-down the interpretation of the study in the conclusion: 'In conclusion, the current data-driven approach using machine learning shows temporal and spatial patterns of activity that serve as a cortical signature for respectively pain, tinnitus, PD, and depression. Our data suggest a spectrally

equivalent but spatially distinct form of TCD depending on the specific neural disorder. However, apart from the disorder-specific spatial signature, common brain areas that are involved in pain, tinnitus, Parkinson's, and depression are identified as well. Therefore, this study supports the existence of TCD as a unifying mechanism underlying diverse neuropsychiatric disorders. However, more research is needed to cross validate these findings, including different neurological and neuropsychiatric disorders.'

- Figure captions should be updated to provide more information (for example in Fig. 5 it is not clear from the figure/caption that the spider plots are showing Pearson correlation coefficients). The authors should also consider labeling the subplots with letters for easier reference.

Answer: We modified the figure captions by adding more information and clarifying the figures more in detail:

'Figure 1. A Comparison of the power spectrum of patients (i.e. tinnitus, pain, PD and depression) with healthy control subjects showed a significant effect for tinnitus ($F = 4.44, p < .001$), pain ($F = 7.77, p < .001$) PD ($F = 3.24, p < .001$), and depression ($F = 3.29, p < .001$) for specific frequencies (see grey bars in figure). A general comparison between all patients (i.e. tinnitus, pain, PD, and depression) and healthy control subjects showed a significant effect ($F = 5.07, p < .001$) for specific frequencies (see grey bars in figure).

Figure 2. Obtained model using support vector machine learning to differentiate between respectively tinnitus vs controls, pain vs controls, Parkinson disease vs controls, and depression vs controls. SVM learning can differentiate between the disorder and healthy control subjects with an accuracy between 75% and 94% in comparison to a random model. The sensitivity of the models and the area under the curve were significantly higher for the obtained model in comparison to the random model, while the false discovery rate was significantly lower. A significant difference was also identified by comparing the κ -statistic MAE and RMSE, confirming the strength of the tested model in comparison to the random model.

Figure 3. Support vector machine learning differentiates between respectively tinnitus vs controls, pain vs controls, Parkinson disease vs controls, and depression vs controls. Abbreviations: dACC: dorsal anterior cingulate cortex; sgACC: subgenual anterior cingulate cortex; INS: Insula; PHC: parahippocampus; AUD: Auditory Cortex, So: Somatosensory cortex; Mo: Motor cortex, PCC: posterior cingulate cortex.

Figure 4. Support vector machine learning differentiates between thalamocortical dysrhythmia as a unifying disorder (including tinnitus, pain, Parkinson's, and depression) vs controls. Abbreviations: dACC: dorsal anterior cingulate cortex; sgACC: subgenual anterior cingulate cortex; INS: Insula; PHC: parahippocampus; AUD: Auditory Cortex, So: Somatosensory cortex; Mo: Motor cortex, PCC: posterior cingulate cortex.

Figure 5. Conjunction analysis between tinnitus, pain, Parkinson's, and depression after the subtraction of the healthy controls shows a significant increase in the dorsal anterior cingulate cortex and hippocampal area for the beta frequency band. dACC: dorsal anterior cingulate cortex; PHC: parahippocampus.

Figure 6. Radar plot of presence of cross-frequency coupling in the auditory cortex, somatosensory cortex, motor cortex, subgenual anterior cingulate cortex, and the dorsal anterior cingulate cortex for theta-beta and theta-gamma coupling using Pearson correlations. An asterisk indicates if the CFC for a specific disorder is significant after Bonferroni correction. The figure demonstrates the presence of theta-gamma (red line) and theta-beta (black line) coupling for tinnitus in the auditory cortex (upper left panel), for pain in the somatosensory (upper right panel) and motor cortices (mid left panel), and Parkinson's in the motor cortex (mid left panel). For the dorsal anterior cingulate (lower left panel) and subgenual anterior cingulate cortices (mid right panel), an increased coupling between theta-

gamma and theta-beta oscillations is identified that is not related to the specific neurological/neuropsychiatric disorder, but likely has a more non-specific general role.

Figure 7. Spatial distribution of theta-beta and theta-gamma cross frequency coupling as related to different thalamocortical dysrhythmia syndromes.'

MINOR POINTS:

- Consider consistently using both “neurological” and “neuropsychiatric” or a more general term such as “neural disorder”.

Answer: We modified this throughout the paper

- There are numerous grammatical mistakes throughout the manuscript – please check carefully.

Answer: We carefully checked for typos and grammatically mistakes.

Reviewer #3

This is an impressive manuscript on a large database of patients similar to the original description of TD in PNAS. I am very supportive of its publication since I do think that TD exists and that the controversy has come more from the technical difficulty in demonstrating it. This manuscript does just that. However, I have a major issue that I believe should be addressed. Since TD is very controversial and this manuscript makes strong claims, I believe that it should not be published without an additional analysis of the EEG similar to the original publication of TD. This analysis should demonstrate that the relevant frequencies are present in the power spectrum of an example patient and the population, show that shifts in frequency are statistically significant and clearly visible in the power spectrum. Another key aspect of TD is the increase in coherence, also illustrated in the original paper, this manuscript must address coherence at the large scale of the EEG.

Answer: We understand the concern of the reviewer and added the power spectrum to the results sections: '*Whole brain frequency analysis*

Comparing the power spectrum of patients (i.e. tinnitus, pain, PD, and depression) with healthy control subjects showed a significant effect for tinnitus ($F = 4.44, p < .001$), pain ($F = 7.77, p < .001$) ($F = 3.29, p < .001$), PD ($F = 3.24, p < .001$), and depression ($F = 3.29, p < .001$). A simple contrast analysis showed a significant increase in the current density for tinnitus patients in comparison to healthy control subjects between 2-4 Hz and 14-44 Hz. In comparison to healthy subjects, we see for pain a significant increase between 2 to 5 Hz and 14 to 44 Hz in current density and a significant decrease between 9 to 10 Hz. For both PD patients and patients with depression, we found respectively a significant increase from 3 to 8 Hz and from 3 and 9 Hz in comparison to healthy control subjects. In addition, a significant increase was identified in current density between 12 and 44 Hz for PD patients and between 19 and 41 Hz for patients with depression in comparison to healthy control subjects. A general comparison between all patients (i.e. tinnitus, pain, PD, and depression) and healthy control subjects showed a significant effect ($F = 5.07, p < .001$). A simple contrast analysis revealed a significant increase between 2 and 5 Hz and between 13 and 44 Hz. See Figure 1 for overview.'

'Figure 1. A Comparison of the power spectrum of patients (i.e. tinnitus, pain, PD, and depression) with healthy control subjects showed a significant effect for tinnitus ($F = 4.44, p < .001$), pain ($F = 7.77, p < .001$) ($F = 3.29, p < .001$), PD ($F = 3.24, p < .001$), and depression ($F = 3.29, p < .001$) for specific frequencies (see grey bars in figure). A general comparison between all patients (i.e. tinnitus, pain,

PD, and depression) and healthy control subjects showed a significant effect ($F = 5.07, p < .001$) for specific frequencies (see grey bars in figure).'

Answer: We also tried to apply an overall cross-frequency coupling (power to power) as applied in the original paper but were unable to find a clear effect looking at power to power. We did not add this to the paper, but if the reviewer thinks this is necessary we are will to add this.

However, we do think that coupling using phase-amplitude gives a more accurate reflection of the relationship between two frequency bands (17) that might be, based on our data that fits with the original paper:

References

1. Volkow ND, *et al.* (2000) Association between age-related decline in brain dopamine activity and impairment in frontal and cingulate metabolism. *The American journal of psychiatry* 157(1):75-80.
2. Logan JM, Sanders AL, Snyder AZ, Morris JC, & Buckner RL (2002) Under-recruitment and nonselective recruitment: dissociable neural mechanisms associated with aging. *Neuron* 33(5):827-840.
3. Siepmann M & Kirch W (2002) Effects of caffeine on topographic quantitative EEG. *Neuropsychobiology* 45(3):161-166.
4. Moazami-Goudarzi M, Michels L, Weisz N, & Jeanmonod D (2010) Temporo-insular enhancement of EEG low and high frequencies in patients with chronic tinnitus. QEEG study of chronic tinnitus patients. *BMC Neurosci* 11:40.
5. Sherlin L & Congedo M (2005) Obsessive-compulsive dimension localized using low-resolution brain electromagnetic tomography (LORETA). *Neurosci Lett* 387(2):72-74.
6. Maki H & Ilmoniemi RJ (2011) Projecting out muscle artifacts from TMS-evoked EEG. *NeuroImage* 54(4):2706-2710.
7. Hernandez-Pavon JC, *et al.* (2012) Uncovering neural independent components from highly artifactual TMS-evoked EEG data. *Journal of neuroscience methods* 209(1):144-157.
8. Liang M, Mouraux A, Chan V, Blakemore C, & Iannetti GD (2010) Functional characterisation of sensory ERPs using probabilistic ICA: effect of stimulus modality and stimulus location. *Clinical neurophysiology : official journal of the International Federation of Clinical Neurophysiology* 121(4):577-587.
9. Litvak V, *et al.* (2007) Artifact correction and source analysis of early electroencephalographic responses evoked by transcranial magnetic stimulation over primary motor cortex. *NeuroImage* 37(1):56-70.
10. Florin E, Pfeifer J, Visser-Vandewalle V, Schnitzler A, & Timmermann L (2016) Parkinson subtype-specific Granger-causal coupling and coherence frequency in the subthalamic area. *Neuroscience* 332:170-180.
11. Buzsaki G & Draguhn A (2004) Neuronal oscillations in cortical networks. *Science* 304(5679):1926-1929.
12. Llinas R, Ribary U, Contreras D, & Pedroarena C (1998) The neuronal basis for consciousness. *Philos Trans R Soc Lond B Biol Sci* 353(1377):1841-1849.
13. Buzsaki G, Logothetis N, & Singer W (2013) Scaling brain size, keeping timing: evolutionary preservation of brain rhythms. *Neuron* 80(3):751-764.
14. Freeman WJ (2006) A cinematographic hypothesis of cortical dynamics in perception. *Int J Psychophysiol* 60(2):149-161.
15. Gray CM & Singer W (1989) Stimulus-specific neuronal oscillations in orientation columns of cat visual cortex. *Proc Natl Acad Sci U S A* 86(5):1698-1702.
16. Llinas RR (1988) The intrinsic electrophysiological properties of mammalian neurons: insights into central nervous system function. *Science* 242(4886):1654-1664.
17. Canolty RT, *et al.* (2006) High gamma power is phase-locked to theta oscillations in human neocortex. *Science* 313(5793):1626-1628.
18. De Ridder D, Vanneste S, Langguth B, & Llinas R (2015) Thalamocortical Dysrhythmia: A Theoretical Update in Tinnitus. *Front Neurol* 6:124.
19. Llinas R, Urbano FJ, Leznik E, Ramirez RR, & van Marle HJ (2005) Rhythmic and dysrhythmic thalamocortical dynamics: GABA systems and the edge effect. *Trends Neurosci* 28(6):325-333.

20. Llinas RR, Ribary U, Jeanmonod D, Kronberg E, & Mitra PP (1999) Thalamocortical dysrhythmia: A neurological and neuropsychiatric syndrome characterized by magnetoencephalography. *Proc Natl Acad Sci U S A* 96(26):15222-15227.
21. Borst A & Theunissen FE (1999) Information theory and neural coding. *Nat Neurosci* 2(11):947-957.
22. Axmacher N, et al. (2010) Cross-frequency coupling supports multi-item working memory in the human hippocampus. *Proc Natl Acad Sci U S A* 107(7):3228-3233.
23. Arnal LH, Wyart V, & Giraud AL (2011) Transitions in neural oscillations reflect prediction errors generated in audiovisual speech. *Nat Neurosci* 14(6):797-801.
24. Song JJ, De Ridder D, Van de Heyning P, & Vanneste S (2012) Mapping tinnitus-related brain activation: an activation-likelihood estimation metaanalysis of PET studies. *Journal of nuclear medicine : official publication, Society of Nuclear Medicine* 53(10):1550-1557.
25. De Ridder D, Elgoyhen AB, Romo R, & Langguth B (2011) Phantom percepts: tinnitus and pain as persisting aversive memory networks. *Proc Natl Acad Sci U S A* 108(20):8075-8080.
26. Vanneste S, et al. (2010) The neural correlates of tinnitus-related distress. *NeuroImage* 52(2):470-480.
27. Vierck CJ, Whitsel BL, Favorov OV, Brown AW, & Tommerdahl M (2013) Role of primary somatosensory cortex in the coding of pain. *Pain* 154(3):334-344.
28. Henson RN, Wakeman DG, Litvak V, & Friston KJ (2011) A Parametric Empirical Bayesian Framework for the EEG/MEG Inverse Problem: Generative Models for Multi-Subject and Multi-Modal Integration. *Frontiers in human neuroscience* 5:76.
29. Mayberg HS (2009) Targeted electrode-based modulation of neural circuits for depression. *The Journal of clinical investigation* 119(4):717-725.
30. Schulman JJ, et al. (2011) Imaging of thalamocortical dysrhythmia in neuropsychiatry. *Front Hum Neurosci* 5:69.
31. Hodkinson DJ, et al. (2016) Increased Amplitude of Thalamocortical Low-Frequency Oscillations in Patients with Migraine. *The Journal of neuroscience : the official journal of the Society for Neuroscience* 36(30):8026-8036.
32. Lauschke JL, Plant GT, & Fraser CL (2016) Visual snow: A thalamocortical dysrhythmia of the visual pathway? *Journal of clinical neuroscience : official journal of the Neurosurgical Society of Australasia* 28:123-127.
33. Schmidt S, et al. (2015) Mindfulness-based Stress Reduction (MBSR) as Treatment for Chronic Back Pain - an Observational Study with Assessment of Thalamocortical Dysrhythmia. *Forschende Komplementarmedizin* 22(5):298-303.
34. Mechias ML, Etkin A, & Kalisch R (2010) A meta-analysis of instructed fear studies: implications for conscious appraisal of threat. *NeuroImage* 49(2):1760-1768.
35. Aminoff EM, Kveraga K, & Bar M (2013) The role of the parahippocampal cortex in cognition. *Trends in cognitive sciences* 17(8):379-390.

Reviewers' comments:

Reviewer #1 (Remarks to the Author):

The authors have adequately addressed my original comments and concerns.

Reviewer #2 (Remarks to the Author):

GENERAL COMMENTS

The study deserves merit for its impressive results. However, details for one of their key analyses, the SVM model, still seem insufficiently characterized for reproducibility, unless perhaps the same particular data-mining software is used. In addition, there are still pervasive careless errors and points that require clarification throughout. More specific points are given below.

SPECIFIC POINTS AND SUGGESTIONS

- Power correlation plots in response to R3 – the authors note that they do not see a clear difference between control and patients samples when looking at power-to-power correlations, but do patients statistically show an increase in theta power, or harmonics in the gamma range? It is difficult to judge from the figure as no key was included. However, these effects were demonstrated in Llinás et al. and would be expected from increased theta coherence and an edge effect respectively, both of which are key arguments for the TCD model of neural disorders.
- Please provide further details on how the SVM model was generated. Were data normalized before being used as features? What kind of kernel was used and what was the slack penalty? These are parameters that may all affect the performance of the SVM and can be optimized via cross-validation, and can affect the interpretability of the results. Experimenting with weighting control and patient classes could also help with the imbalanced class sizes.
- Additionally, could the authors please clarify what is meant by a “simple logistics filter” (line 214)?
- “False discovery rate” and “FPR” (false positive rate) have different statistical definitions, but they appear to be used interchangeably throughout this manuscript. Please clarify.
- Fig. 3 – it is still unclear to me why frequency bands are only listed for some regions and not others here, and this information is not included in the caption. A fellow reviewer also brought up this point, which I feel is still inadequately addressed.

MINOR POINTS

- The manuscript is now much easier to read and understand than it had been previously, but there are still several typos throughout – e.g. in the Abstract, “Parkinson’s Disease (PD)” should be “Parkinson’s Disease (PD)”; line 284, “modal” should be “model” etc.
- There are also some abbreviations that are not defined in the text or in the captions (e.g. TPR, FPR).
- In Fig. 2 “Percentage” is used as the y-axis for plots in the first column but decimals are used for the second and third columns – please consider keeping the labels consistent.

Reviewer #3 (Remarks to the Author):

I am impressed and convinced by the additional analysis, I hope the authors agree that this increases the power of the results. I don't think that the figure illustrating the lack of cross frequency coupling is necessary, but it should be cited and the phase-amplitude analysis should be in the manuscript and explained as in the reply to the reviewer. otherwise i have no further comments. i enjoyed reading both versions of the manuscript.

Response to the Reviewer

Reviewer #2

GENERAL COMMENTS

The study deserves merit for its impressive results. However, details for one of their key analyses, the SVM model, still seem insufficiently characterized for reproducibility, unless perhaps the same particular data-mining software is used. In addition, there are still pervasive careless errors and points that require clarification throughout. More specific points are given below.

SPECIFIC POINTS AND SUGGESTIONS

- Power correlation plots in response to R3 – the authors note that they do not see a clear difference between control and patients samples when looking at power-to-power correlations, but do patients statistically show an increase in theta power, or harmonics in the gamma range? It is difficult to judge from the figure as no key was included. However, these effects were demonstrated in Llinás et al. and would be expected from increased theta coherence and an edge effect respectively, both of which are key arguments for the TCD model of neural disorders.

Answer: We do see an increase in the theta and gamma power for our patients (see figure 1). However, for the power to power cross-frequency coupling we did not identify a significant difference between controls and patients. More recent research suggests that phase-amplitude is a more accurate reflection of the relationship between the two frequencies¹. Applying this latter method demonstrates a clear correlation between the phase of the theta and the amplitude of the gamma.

- Please provide further details on how the SVM model was generated. Were data normalized before being used as features? What kind of kernel was used and what was the slack penalty? These are parameters that may all affect the performance of the SVM and can be optimized via cross-validation, and can affect the interpretability of the results. Experimenting with weighting control and patient classes could also help with the imbalanced class sizes.

Answer: We understand the concern. Therefore, we rewrote the whole section to clarify the method further. In addition, we added more information in the supplementary materials regarding the method applied. To further analyze the problem of the imbalanced class sizes we include also weighted comparisons.

- Additionally, could the authors please clarify what is meant by a “simple logistics filter” (line 214)?

Answer: We modified this in the paper and changed it to logistic regression-based classifier. We further added in supplementary material the underlying method.

- “False discovery rate” and “FPR” (false positive rate) have different statistical definitions, but they appear to be used interchangeably throughout this manuscript. Please clarify.

Answer: Was a typo. We change if throughout the paper to false positive rate.

- Fig. 3 – it is still unclear to me why frequency bands are only listed for some regions and not others here, and this information is not included in the caption. A fellow reviewer also brought up this point, which I feel is still inadequately addressed.

Answer: We understand the confusion and regenerated the figures only including the areas that show up based on the SVM calculations and not the areas that were included in the SVM model. We did the same thing for figure 4 to avoid inconsistency.

MINOR POINTS

- The manuscript is now much easier to read and understand than it had been previously, but there are still several typos throughout – e.g. in the Abstract, “Parkinson’s Disease (PD)” should be “Parkinson’s Disease (PD)”; line 284, “modal” should be “model” etc.

Answer: We corrected this.

- There are also some abbreviations that are not defined in the text or in the captions (e.g. TPR, FPR).

Answer: We screened the whole paper and added the missing abbreviations.

- In Fig. 2 “Percentage” is used as the y-axis for plots in the first column but decimals are used for the second and third columns – please consider keeping the labels consistent.

Answer: We modified the figure so that it is consistent with the second and third columns.

Reviewer #3

I am impressed and convinced by the additional analysis, I hope the authors agree that this increases the power of the results. I don't think that the figure illustrating the lack of cross frequency coupling is necessary, but it should be cited and the phase-amplitude analysis should be in the manuscript and explained as in the reply to the reviewer.

Answer: We understand the concern of the reviewer and integrated the comment as suggested. ‘Phase–amplitude was chosen over power–power cross-frequency coupling as the former has been shown to reflect a physiological mechanism for effective communication in the human brain¹.’

1 Canolty, R. T. *et al.* High gamma power is phase-locked to theta oscillations in human neocortex. *Science* **313**, 1626-1628, doi:10.1126/science.1128115 (2006).

REVIEWERS' COMMENTS:

Reviewer #3 (Remarks to the Author):

I have no further general comments except that I believe the significant increase in the theta and gamma power in patients shown in figure 1 without a significant difference in the power to power cross-frequency coupling MUST be clearly indicated in the results section since this differs from the original PNAS report on TCD. It would also be highly appropriate and useful if the authors included a comment in discussion commenting on the recent research that suggests that phase-amplitude is a more accurate reflection of the relationship between the two frequencies, with citations.

Response to the editor reviewers:

Reviewer #3 (Remarks to the Author):

I have no further general comments except that I believe the significant increase in the theta and gamma power in patients shown in figure 1 without a significant difference in the power to power cross-frequency coupling MUST be clearly indicated in the results section since this differs from the original PNAS report on TCD. It would also be highly appropriate and useful if the authors included a comment in discussion commenting on the recent research that suggests that phase-amplitude is a more accurate reflection of the relationship between the two frequencies, with citations.

Answer: We understand the concern of the reviewer and add the cross-frequency coupling in the results section and include a comment in the discussion.

“The cross-correlation between spectral amplitudes at different frequencies for healthy control subjects as well as for patients with tinnitus, pain, Parkinson’s disease, and depression as illustrated in Fig. 7 does not show a significant difference in increased power to power in theta-beta and theta-gamma correlation between the healthy control group and the patient groups.”

“Theta-beta and theta-gamma coupling were however not confirmed when using a power-to-power cross-frequency coupling analysis as applied in the original TCD model⁹. However, more recent research suggests that phase–amplitude coupling more accurately reflects the physiological mechanism for effective communication in the human brain⁶.”